

# Phytoplankton and dimethylsulfide dynamics at two contrasting Arctic ice edges

Martine Lizotte[1], Maurice Levasseur[1], Virginie Galindo[2], Margaux Gourdal[1], Michel Gosselin[2], Jean-Éric Tremblay[1], Marjolaine Blais[3], Joannie Charette[4], Rachel Hussherr[1]

[1]Département de biologie, Québec-Océan, Université Laval, Québec, Québec, G1V 0A6, Canada

[2]Institut des sciences de la mer de Rimouski (ISMER), Université du Québec à Rimouski, Rimouski, Québec, G5L 3A1, Canada

[3]Maurice Lamontagne Institute, Fisheries and Oceans Canada, Mont-Joli, Québec, G0J 2L0, Canada

[4]Fisheries and Oceans Canada, Winnipeg, Manitoba, R3T 2N6, Canada

*Correspondence to*: Martine Lizotte (martine.lizotte@qo.ulaval.ca)

**Abstract.** Arctic sea ice is retreating, thinning and its rate of decline has steepened in the last decades. While phytoplankton blooms are known to seasonally propagate along the ice edge as it recedes from spring to summer, the substitution of thick multi-year ice (MYI) with thinner, ponded first-year ice (FYI) represents an unequal exchange when considering the roles sea ice plays in the ecology and climate of the Arctic. Consequences of this shifting sea ice on the phenology of phytoplankton and the associated cycling of the climate-relevant gas dimethylsulfide (DMS) and its precursor dimethylsulfoniopropionate (DMSP) remain ill constrained. In July-August 2014, two contrasting ice edges in the Canadian High Arctic were explored: a FYI-dominated ice edge in Barrow Strait and a MYI-dominated ice edge in Nares Strait. Our results reveal two distinct planktonic systems and associated DMS dynamics in connection to these diverging ice types. The surface waters exiting the ponded FYI in Barrow Strait were characterized by moderate chlorophyll *a* (Chl *a*, < 2.1 µg L$^{-1}$) as well as high DMSP (115 nmol L$^{-1}$) and DMS (12 nmol L$^{-1}$) suggesting that a bloom had already started to develop under the markedly melt pond-covered (ca. 40%) FYI. Heightened DMS concentrations at the FYI edge were strongly related with ice-associated seeding of DMS in surface waters and haline-driven stratification linked to ice melt (Spearman's rank correlation between DMS and salinity, $r_s$ = -0.91, p < 0.001, n = 20). However, surface waters exiting the MYI edge at the head of Nares Strait were characterized by low concentrations of Chl *a* (< 0.5 µg L$^{-1}$), DMSP (< 16 nmol L$^{-1}$) and DMS (< 0.4 nmol L$^{-1}$), despite the nutrient-replete conditions characterizing the surface waters. The increase in autotrophic biomass and methylated sulfur compounds took place several km (ca. 100 km) away from the MYI ice edge suggesting the requisite for ice-free, light-sufficient conditions for a phytoplankton bloom to fully develop and for sulfur compound dynamics to follow and expand. In light of the ongoing and projected climate-driven changes to Arctic sea ice, results from this study suggest that the early onset of autotrophic blooms under thinner, melt pond-covered ice may have vast implications for the timing and magnitude of DMS pulses in the Arctic.



## 1 Introduction

The rapid warming of the Arctic represents one of the most conspicuous impacts of global change driven by human activities (IPCC 2013). This warming has already translated into widespread and profound modifications in hydrological and ecological systems including, but not limited to, those related to sea ice dynamics. Reductions in snow cover, in the minimum sea ice cover in summer and in the occurrence of multi-year (MYI) ice are afoot and hastening (Rothrock et al. 1999; Serreze et al. 2007; Stroeve et al. 2007; 2008; Comiso et al. 2008; Serreze and Stroeve 2015; Bockhorst et al. 2016; AMAP 2017). If warming continues unmitigated, a summer ice-free Arctic Ocean is predicted to occur during the second half of this century (Wang and Overland 2012). In polar regions, primary production is driven by light and nutrient availability (Loeng et al. 2005; Arrigo 2014) which are heavily influenced by the presence of sea ice. Seasonally, the first autotrophic organisms to benefit from the vernal increase in light in the Arctic are ice algae that develop mainly in the bottom ca. 2-5 cm of the ice (Gradinger 2009; Galindo et al. 2014; van Leeuwe et al. 2018). During this period, light intensities under thick ice are commonly too low to allow phytoplankton growth, however blooms may develop in ice-free waters in long narrow bands (20-100 km) trailing along the ice edge (Sakshaug and Skjoldal 1989). These ice-edge blooms could account for up to 50% of the annual primary production in the Arctic Ocean (Perrette et al. 2011) and represent the greatest supply of energy to the marine Arctic ecosystem (Wassmann et al. 2008). Later in the growing season, phytoplankton blooms can also develop under the ice as light penetration through the ice pack is heightened due to snow melting, ice thinning, and the development of melt ponds (Fortier et al. 2002; Mundy et al. 2007; Arrigo et al. 2012; Galindo et al. 2014). The gradual loss of perennial sea ice, induced by climate change, is thus expected to influence the phenology of ice-edge blooms and the associated biogeochemical cycling of elements such as carbon, nitrogen, and sulfur. Based on data from an under-ice bloom from the Chukchi Sea (Arrigo et al. 2012), a modelling study by Palmer et al. (2014) shows that a 10% melt pond cover at the surface of the ice pack could provide sufficient light to sustain the growth of shade-adapted algae under the ice. While conditions required for under-ice blooms were likely scarcer no more than 20-years ago, another modelling study found that nearly 30% of the ice-covered Arctic Ocean may now support the formation of under-ice blooms during the month of July (Horvat et al. 2017).

Ice-covered seas not only shape marine food webs but they also influence ocean-atmosphere exchanges of energy, particles and gases, including the climate-cooling compound dimethylsulfide (DMS) (Levasseur 2013; Gabric et al. 2018). In the remote marine atmosphere, ocean-originating DMS represents the greatest gaseous precursor of sulfur containing aerosols (Bates et al. 1992; Andreae and Crutzen 1997). Sulfate aerosols may play an important role in the Earth's radiative budget as they scatter incoming shortwave radiation and influence cloud formation and precipitation by operating as cloud condensation nuclei (CCN) (Andreae 1990; Curran and Jones 2000; Liss and Lovelock 2007). The potential importance of oceanic DMS emissions in driving climate-cooling is greatest in regions characterized by low burdens of airborne particulates such as in the Arctic during summer (Chang et al. 2011; Browse et al. 2012; Carslaw et al. 2013; Leaitch et al. 2013).

DMS rises in great part from the degradation of the algal compound dimethylsulfoniopropionate (DMSP). DMSP holds several roles in unicellular algae including osmoregulation, cryoprotection, scavenging of free radicals, and overflow of carbon and sulfur (Stefels et al. 2007). The production of DMSP by unicellular algae is highly species-specific with Bacillariophyceae and Dinophyceae/Prymnesiophyceae being lesser and greater producers, respectively (Keller et al. 1989). The DMSP-to-DMS conversion involves the entire microbial food web and part of the DMS is produced directly by phytoplankton while another part is produced indirectly via the release of DMSP in the aqueous environment and its subsequent degradation by bacterioplankton (Kiene et al. 2000; Simó 2001; Stefels et al. 2007). The relative importance of



these processes is unclear, however abiotic stressors involving sudden modifications in light intensity, salinity, and temperature may all contribute to the enhanced direct and indirect production of DMS by plankton communities (Sunda et al. 2002; Toole and Siegel 2004).

In the Arctic, peaks in atmospheric methane sulfonic acid (MSA, a DMS proxy) have frequently been measured in spring and in mid-summer (Sharma et al. 2012). The spring peaks have been attributed to phytoplankton blooms at low latitudes

while the mid-summer peaks have been related to more localized high latitude ice edge blooms (Sharma et al. 2012; Becagli et al. 2016; 2019). This interpretation is consistent with the elevated DMS concentrations generally measured at or close to ice edges in association with developing phytoplankton blooms in the North Atlantic and European sectors of the Arctic (Matrai and Vernet 1997; Galí and Simó 2010; Park et al. 2018). The high DMS concentrations measured at ice edges have been associated with a combination of factors including: 1) an increase in phytoplankton biomass and hence DMSP

concentrations; 2) the selective growth of strong DMSP and DMS producers such as the prymnesiophyceae *Phaeocystis*; 3) a physiological stimulation of DMS production due to the increase in irradiance; and 4) an increase in bacterial activity (Galí and Simó 2010). In the eastern Canadian High Arctic, only a fragmented picture of summer oceanic DMS distributions was available until recently and none of the snapshots captured the presumably most biologically productive time of July-August: average of 1.1 nmol DMS $L^{-1}$ in the North Water and Nares Strait in June (Bouillon et al. 2002); average of 1.3 nmol DMS $L^{-1}$

in northern Baffin Bay/Lancaster Sound in September (Motard-Côté et al. 2012); range of 0.05 to 0.8 nmol DMS $L^{-1}$ in the Canadian High Arctic in October/November (Luce et al. 2011). In spite of the recurring mid-summer atmospheric MSA peak measured at Alert, evidence of high oceanic DMS concentrations associated with summer phytoplankton blooms remained scarce for this part of the Arctic until very recently (Mungall et al. 2016; Collins et al. 2017; Jarníková et al. 2018; Abbatt et al. 2019).

The rapid shifting of the Arctic icescape bears consequences for Arctic primary production and associated DMS dynamics that are still poorly understood. While observations from the field are sparse (Wassmann et al. 2011) and challenging due the remoteness and harshness of the environment as well as the dynamic nature of ice and its margins (Sakshaug and Skjoldal 1989), it is critical that impacts of ongoing physical changes on the dynamics of bloom-forming microorganisms and their production of DMS be better constrained. The main objective of this study was to assess and compare mid-summer (July-

August) phytoplankton and DMS dynamics at two contrasting ice edges in regions of the eastern Canadian Arctic: the Barrow Strait first-year ice (FYI) dominated ice edge and the Nares Strait multi-year ice (MYI) dominated ice edge. The opportunity was also taken to investigate the ice-free waters of Lancaster Sound and North Water (northern Baffin Bay) contiguous to the Barrow Strait and Nares Strait ice edge regions, respectively. Our results reveal two distinct planktonic systems and ensuing DMS dynamics related to the presence of dissimilar icescapes.


## 2 Methods

### 2.1 Ice conditions and sampling strategy

The sampling took place between July 17 and August 6, 2014 onboard the Canadian Coast Guard Ship (CCGS) Amundsen as part of the joint ArcticNet/NETCARE (Network on Climate and Aerosols: Addressing Key Uncertainties in Remote

Canadian Environments) campaign. Thirty-three stations were sampled; most of them were located on four strategic and historical ArcticNet transects: Barrow Strait, Lancaster Sound, Nares Strait and North Water (Table 1, Fig. 1). The other stations were situated in Dease Strait, Victoria Strait, M'Clintock Channel and Franklin Strait in the CAA and Baffin Bay.





The Barrow Strait (BS) transect was sampled opportunistically and aimed to capture seawater flowing eastward as it exited the ponded ice pack in Barrow Strait. The Lancaster Sound (LS) transect captured the water masses coming in and out of the

Sound. The Nares Strait (NS) transect aimed at capturing the progression of biochemical processes as the water flowed southward away from the northern ice arch. The North Water (NOW) transect captured the exchanges between the northern part of Baffin Bay and Nares Strait.

The two Straits (Barrow and Nares) were characterized by distinct and well-defined ice edges at the time of sampling (Fig. 2). In Barrow Strait, the ice edge was located at the western end of Lancaster Sound, perpendicular to the channel, between

Devon Island and Somerset Island (Fig. 2a). The ice pack was mostly composed of ca. 1 m thick FYI covered by melt ponds at approximately 40% of total surface (Fig. 3 picture of melt ponds). Soon after our arrival in the study area, a large lead developed south of Griffith Island (south of Cornwallis Island), pushing the detached part of the ice pack slightly eastward (Fig. 2). The BS transect was conducted along the ice edge in this lead. In Barrow Strait, the net surface circulation is predominantly eastward at 10-15 cm s$^{-1}$ in mid-summer on the south shore with a mild westward current of ca. 5 cm s$^{-1}$ on

the north shore (Lemon and Fissel 1982; Prinsenberg and Bennett 1987; Pettipas et al. 2008, Michel et al. 2015). This region stands as an important waterway for the transport of fresher Pacific waters, originally from the inflow through Bering Strait, towards the North Atlantic (Jones et al. 2003). The water sampled across this transect was thus mostly exiting the ice pack which extended several km westwards.

In July 2014, an ice arch formed in the Kennedy Channel of Nares Strait leaving Kane Basin, and the North Water region to

the south largely ice-free. The comparison of the position of the ice arch in July 2014 with a decade of remotely sensed data (1997-2007), shows that it formed that year approximately 130 km north of a median historical position (near 79$^{o}$N) in southern Kane Basin (Kwok et al. 2010), in line with recent trends (2006-2010) of more northern ice bridge formation in the area (Ryan and Münchow 2017). By the time of the sampling (3-6 August), it had retreated to the head of Kennedy Channel (Fig. 2A), leaving a 350 km stretch of open water north of Smith Sound (Burgers et al. 2017). As expected for this part of

the Arctic Ocean, the ice pack north of the ice arch was composed of MYI (Fig. 2C). Presence of MYI (5+ years) north of Nares Strait, near Robeson Channel, was confirmed by the Ease-Grid Sea Ice Age, Version 3 data set (Tschudi et al. 2016), which compiles weekly estimates of sea ice age in the Arctic between 1978 and 2017. Data from 2014, week 31 (28 July–3 August) and week 32 (4–10 August) were consulted for the purpose of this study. Beyond the MYI and to the south, a band of thick (>1.2 m) FYI, without any melt ponds, was also present (Canadian Ice Service (CIS) analysis, Fig. 2B). Because

Nares Strait represents a major outflow path for water exiting the Arctic Ocean (Jones et al. 2003; Münchow et al. 2007; McGeehan and Maslowski 2012), the water sampled along the NS transect was exiting the northern MYI edge as it flowed southbound towards Baffin Bay.

### 2.2 Physical, chemical and biological measurements

Water samples were collected at 5 to 9 depths from the surface down to a maximum of 100 m depth with 12-L Niskin-type bottles mounted on a General Oceanics 24-bottle rosette. The rosette sampler was equipped with a Sea-Bird 911plus Conductivity Temperature Depth (CTD) probe and a sensor for the measurement of fluorescence (Seapoint). The *in vivo* fluorescence data were calibrated against extracted chl *a* concentration. Data processing was performed through the "Sea-Bird SBE Data Processing program", and quality control (based on UNESCO algorithms) was performed using a Matlab toolbox

developed at the Maurice Lamontagne Institute (Fisheries and Ocean Canada) (Guillot 2007, unpub.). Water for nutrient



analysis was filtered using a luer-lock syringe combined with an Acrodisc filter (0.7 µm) into 15-mL acid-washed polyethylene tubes. Samples were immediately analyzed for nitrate ($NO_3^-$), nitrite ($NO_2^-$), phosphate ($PO_4^{3-}$) and silicic acid ($Si(OH)_4$) using a Bran and Luebbe Autoanalyser III after the colorimetric method adapted from Hansen and Koroleff (1999) (detection limit for $NO_3^-$: 0.03 µmol $L^{-1}$, $NO_2^-$: 0.02 µmol $L^{-1}$, $PO_4^{3-}$: 0.05 µmol $L^{-1}$).

Water for chlorophyll $a$ (chl $a$) concentration analysis was collected in 1-L brown polyethylene bottles (Nalgene) and then passed onto 25-mm filters (Whatman GF/F). Phytoplankton pigments were extracted in 90% acetone and stored at 4ºC in the dark during a period of 18-24 hours. Fluorescence of extracted pigments was then measured using a Turner Designs fluorometer 10-AU after the acidification method described by Parsons et al. (1984). Chl $a$ concentrations were calculated from the equation published in Holm-Hansen and collaborators (1965).

Samples for phytoplankton taxonomy were collected at the surface and at the subsurface chlorophyll maximum (SCM) and preserved in an acidic Lugol's solution (final concentration of 0.4% v:v; Parsons et al. 1984). Identification and enumeration of cells > 2 µm were conducted with a Zeiss Axiovert 10 inverted microscope following the Utermöhl and Lund method (Lund et al. 1958; Parsons et al. 1984). A minimum of 400 cells was enumerated to be statistically significant.

Samples of DMS were collected in 23-ml serum vials and allowed to gently overflow, avoiding any bubbling, before capping.

Concentrations of DMS were determined onboard within 2 hours of collection using purging, cryotrapping, and sulfur-specific gas chromatography (GC, Varian 3800) as described by Lizotte et al. (2012) and further modifications described here. Briefly, 15 to 20-ml subsamples of DMS were gently filtered through a GF/F syringe filter and immediately injected into a sparging vessel. The DMS was stripped from the liquid samples using a constant flow of Ultra High Purity (UHP) helium (He) prepared using a permeation tube (certified calibration by Kin-Tek Laboratories Inc.) maintained at 40ºC and

volatile DMS was trapped in a Teflon loop held in liquid $N_2$. Gaseous samples were then analyzed using a Varian 3800 gas chromatograph (GC), equipped with a Pulsed Flame Photometric Detector (PFPD). The samples were calibrated against microliter injections of DMS diluted with UHP He (certified calibration by Kin-Tek Laboratories Inc.) maintained at 40ºC. Duplicate tubes for total DMSP ($DMSP_t$) samples were filled with 3.5 mL of unfiltered water. For conservation purposes, 50 µL of 50% sulfuric acid ($H_2SO_4$) was added in each 3.5 mL liquid sample of $DMSP_t$. All tubes were stored at 4ºC in the

dark until analysis in laboratory. DMSP concentrations were quantified over the course of two periods using two analytical systems. A first series of $DMSP_t$ samples (stations 323, 322, 325, 301, 304, 305, 305A, 305B, 305C, 305D, and 305E) was analyzed in the laboratories of Laval University using a purge and trap system coupled to a Varian 3800 GC PFPD as described above. $DMSP_t$ samples were hydrolyzed with a 5N NaOH solution in order to convert DMSP into DMS which was purged from the samples via an Ultra High Purity (UHP) helium stream, cryo-trapped and analyzed via gas

chromatography (Lizotte et al. 2012). For these DMSP samples, the GC was calibrated with milliliter injections of a 100 nmol $L^{-1}$ solution of hydrolyzed DMSP (Research Plus Inc.).

The analytical detection limit on the Varian GC system was 0.1 nmol $L^{-1}$ for all sulfur compounds and the analytical precision (CV) for triplicate measurements of DMS and DMSP was better than 10%. After shortcomings with the aforementioned GC system, a second series of $DMSP_t$ samples (stations 300, 324, 346, 115, 111, 108, 105, 101, KEN1, KEN3, KANE1, KANE3,

314, 312, 310, 335, 210, 204, 200, and 120) were determined using an automated purge and trap system (Atomx XYZ, Teledyne Tekmar Inc.) coupled with a GC-MS (Gas chromatograph - Mass Spectrometer, model GC Intuvo 9000-MS 5977B, Agilent Inc.). Before analysis on the GC-MS, DMS derived from NaOH-hydrolyzed $DMSP_t$ samples was purged from the seawater in the Atomx sparge vessel for 11 minutes with UHP He at a flow rate of 40 mL $min^{-1}$. Purged DMS was trapped





on a u-shaped proprietary trap. A high-voltage current was then used to heat the trap to 250°C, desorbing DMS and sending

it to the elution column of the GC at a flow rate of 300 mL min$^{-1}$. Once separated by the GC column, volatile compounds were ionized and directed to the mass selective quadrupole of the MS. The detector was adjusted to count target ions at 62 *m/z*. Resulting peak areas were retrieved using the MassHunter workstation software. DMSP concentrations were calculated against 6-7 point calibration curves obtained by processing standard DMSP solutions of known concentration, between 1 and 100 nmol L$^{-1}$, in the same fashion as DMSP samples. Potential degradation of DMSP$_t$ samples through time was corrected

by calculating the loss in standard solutions of DMSP (100 nmol L$^{-1}$) kept in the same preservation conditions as the samples (4°C in the dark). Our analysis shows an average loss of 9% in the DMSP$_t$ samples between times of sampling and analysis. MODIS images, as well as ice charts produced by CIS, were used to visually assess the presence of ice edges. CIS ice charts, based on Radarsat 2 and NOAA-18 images, show ice properties including stage of development, concentration and form of the ice (Environment Canada 2005). Color schemes of the CIS ice chart were modified using Adobe Illustrator CS6. A FYI

edge appears in Lancaster Sound as a curved line between Devon Island and Somerset Island on July 22 (Fig. 2A). The presence of MYI appears at the northern extremity of Nares Strait, i.e., at the entrance of Robeson Channel between Ellesmere Island and Greenland, on August 1 (Fig. 2C). The MYI ice was contiguous to a band of thick (> 1.2 m) FYI descending into Nares Strait (Fig. 2B).

The surface mixed layer depth ($Z_m$) was estimated as the depth at which the gradient in density ($\sigma_t$) between two successive

depths was greater than 0.03 kg m$^{-4}$ following the threshold gradient method of Thomson and Fine (2003) with adaptations from Tremblay et al. (2009). Oceanic vertical cross sections and contour plots were drawn using weighted averaging gridding and linear mapping using Ocean Data View 5.1.5 macx software (Schlitzer, 2018) and schematic models of FYI and MYI dynamics were constructed in Adobe Illustrator CS6. Statistical analysis was conducted using SYSTAT 13.2 software, as well as JASP 0.9.2.0 computer software, an open-source project supported by the University of Amsterdam (JASP Team

2018). Variables were tested for normality using the Shapiro-Wilk test with a 0.05 significance level, and Spearman's rank correlations ($r_s$) were used to assess the strength of association between variables.

### 3 Results

### 3.1 Overview of the sea surface physicochemical and biological characteristics

The main physical and chemical characteristics of the sea surface water at the sampling stations are presented in Table 1 for stations located in the 4 regions of interest. Surface temperature varied between -1.5 and 5.7°C, with the lowest and highest values measured under the ice in Barrow Strait and in the open waters of Lancaster Sound, respectively. Salinities ranged from 29.5 to 32.8, with the lowest and highest values measured in Barrow Strait and in northern Baffin Bay (North Water), respectively. Nitrate concentrations were generally lower than 0.5 µmol L$^{-1}$ in the studied area, except at three ice edge

stations located in Barrow Strait (> 2 µmol L$^{-1}$) and one station close to the ice arch in Nares Strait (1.4 µmol L$^{-1}$). Silicic acid exhibited the same general spatial distribution with concentrations lower than 1.5 µmol L$^{-1}$ at most stations and greater than 3 µmol L$^{-1}$ in the ice-covered Barrow Strait and close to the ice arch in Nares Strait. Phosphate concentrations varied between 0.2 and 0.9 µmol L$^{-1}$, again with maximum values found in ice-covered Barrow Strait. Chlorophyll *a* concentrations in surface waters varied between 0.2 and 2.3 µg L$^{-1}$ (Table 2), indicating that the summer bloom was in an advanced stage at

most stations, except in the ice-covered Barrow Strait and close to the ice arch in Nares Strait. Surface DMSP$_t$ concentrations spanned an order magnitude from 13.5 to 114.7 nmol L$^{-1}$, while DMS concentrations varied between 0.37 and 19.5 nmol L$^{-1}$,



with the highest values measured in the ice-covered Barrow Strait and near the Greenland shelf in northern Baffin Bay (Table 2). At a broad scale, and considering only sea surface data from all regions under investigation in this study, Spearman's rank correlation tests (n = 33) reveal no significant relationships between DMS and abiotic or biotic variables presented in tables 2 and 3.

Beyond sea surface data, water column vertical profiles were also plotted as cross sections in order to identify key features associated with ice dynamics and bloom development in certain regions of the CAA and Baffin Bay. Information is presented below and grouped as a function of targeted transects.

**3.2 Barrow Strait (BS) transect**

Variables measured across the BS transect are presented in Figure 4. Seawater temperatures ranged from -1.6 to -1.2ºC, with the lowest values found at intermediate depths (ca. 40-60 m). Surface water temperatures were below -1.4°C at all stations. Salinity varied between 30.4 and 33.0 across the transect, with the lowest and highest surface values measured at the north and south extremities of the transect, respectively. Nitrate concentrations ranged from 0.6 to 11.0 µmol $L^{-1}$, with lowest and

highest values measured close to the surface and at depth, respectively. The nitracline was located at ca. 30 m. Close to the surface, nitrate concentrations were low at the south end of the transect (0.6 µmol $L^{-1}$ at station 305B) and increased northward to reach 2.1 µmol $L^{-1}$ at station 305E. Silicic acid concentrations showed a similar pattern, with a positive south-north gradient ranging from 3.5 to 10.5 µmol $L^{-1}$ in the upper 30 m and high values at depth (up to 29.2 µmol $L^{-1}$). Chl *a* concentrations varied between 0.2 and 2.1 µg $L^{-1}$ with highest values measured in the upper 30 m of the water column and

toward the northern tip of the transect. Phytoplankton identification and enumeration were conducted at one station on the BS transect (stations 305E) and at two stations located in the vicinity under the ponded ice cover (see stations 304 and 305 in Fig. 1 and Table 2). The phytoplankton assemblages at these three stations were similar, dominated by the pennate diatoms *Fossula arctica* and *Pseudo-nitzschia* spp. (*delicatissima* group)*, the two taxa being responsible for 29 to 71% of the total phytoplankton abundance (Table 2). Another abundant pennate species at these stations was *Fragilariopsis oceanica.*

Concentrations of $DMSP_t$ were highest in the top 20-30 m of the water column across the FYI edge, from stations 305B to 305E, with the highest value of $DMSP_t$ (115 nmol $L^{-1}$ at 2 m depth) observed at the northern extremity of the transect. DMS concentrations were maximal in the upper 30 m of the water column across the BS transect. DMS concentrations in surface waters varied from 7.2 to 12.2 nmol $L^{-1}$ with highest concentrations measured at both extremities of the transect.

**3.3 Lancaster Sound (LS) transect**

Variables measured across the LS transect are presented in Figure 5. Surface temperatures were at least 3 times warmer than those measured across the BS transect, with values ranging between 3.0 and 4.1ºC. Surface salinities varied between 30.7 and 32.4, with the highest values measured at stations 323 and 322 towards the north shore. Concentrations of nitrate and silicic acid exhibited no particular cross-channel pattern in the surface mixed layer, with values below 0.5 and 2 µmol $L^{-1}$ in

the upper 20 m of the water column, respectively. Maximum Chl *a* concentrations were in the same range as in the BS transect (between 1.5 and 2.5 µg $L^{-1}$) but exhibited a different vertical distribution. Across the BS transect, Chl *a* concentrations were generally highest in the surface mixed layer (SML) while they formed a SCM at ca. 30-40 m at the stations located across the LS transect suggesting a more advanced bloom stage in the LS area. The two transects also showed distinct phytoplankton assemblages (Table 2). Station 325 located close to the south shore of the LS transect was



characterized by a phytoplankton assemblage dominated by the centric diatoms *Chaetoceros* spp. (5-10 µm), *Chaetoceros gelidus*, *Chaetoceros* spp. (10-20 µm) and *Chaetoceros* spp. (2-5 µm). At stations located in the middle (323) and near the north shore (322), assemblages were dominated by unidentified flagellates, Prasinophyceae and Dinophyceae. In contrast with assemblages found at the BS transect, pennate diatoms represented at most 7% of the phytoplankton counts in Lancaster Sound.

Concentrations of DMSP$_t$ varied between 17 and 96 nmol L$^{-1}$ in the top 40 m, with a pronounced subsurface peak at ca. 20 m at the southern end of the transect corresponding to the SCM. DMS concentrations were as high across the LS transect (values above 10 nmol L$^{-1}$ measured at stations 322, 300, and 325) than those measured across the BS transect. However, while the high DMS values were restricted to the first 25 m in the BS transect, concentrations exceeding 2.5 nmol L$^{-1}$ were measured down to ca. 70 m in the LS transect. Across the LS transect, DMS concentrations were elevated (> 4 nmol L$^{-1}$) in the nutrient-

impoverished low Chl *a* SML as well as in the SCM. Three distinctive peaks were observed at stations 322 (10 nmol L$^{-1}$ at 40 m), 322 (12 nmol L$^{-1}$ at 20 m), and 325 (11 nmol L$^{-1}$ at 10 m).

### 3.4 Nares Strait (NS) transect

Variables measured across the NS transect are presented in Figure 6. Sea surface temperatures started at ca. -1.3°C at the ice

edge and increased more or less regularly southward to reach 2°C at the last station (KANE5) of the transect. In contrast, sea surface salinities were relatively constant at 30.5 along the transect. Nitrate and silicic acid concentrations in surface waters near the ice arch were ca. 1.5 µmol L$^{-1}$ and 6 µmol L$^{-1}$, respectively. In the upper 20 m of the water column, concentrations of nitrate and silicic acid decreased with distance from the ice arch as a first algal bloom developed (see below), reaching 0.4 µmol L$^{-1}$ and 1.9 µmol L$^{-1}$, respectively, at the southernmost station (KANE5). The silicic acid drawdown along the

transect was indicative of a strong diatom dominance (see Table 2).

Chl *a* concentrations exiting the MYI pack were ca. 0.3 µg L$^{-1}$ in the top 20 m of the water column and increased southward, reaching a first surface peak of 2 µg L$^{-1}$ at KEN3 which then continued in subsurface waters. A SCM of 2.8 µg L$^{-1}$ was already present at ca. 24 m depth at KEN3, while Chl *a* concentrations reached 10 µg L$^{-1}$ at ca. 20 m depth at KANE5. The abundance of phytoplankton was low in surface waters near the MYI edge and unidentified flagellates and Prymnesiophyceae

(Table 2) dominated the community. The bloom which developed further south of the ice arch was dominated by the centric diatoms *Chaetoceros* spp. (5-20 µm) and *Chaetoceros gelidus*, a composition similar to the blooming assemblage described in the LS transect.

At the northernmost station near the ice arch (KEN1), DMSP$_t$ concentrations were relatively low throughout the water column (< 16 nmol L$^{-1}$), with highest values near the surface. The near surface maximum increased to 27 nmol L$^{-1}$ at station

KEN3 while a distinct subsurface maximum of DMSP$_t$ was present at ca. 20m depth in the three southernmost stations of the transect (KANE1 to KANE5). A high value of 59 nmol L$^{-1}$ was reached at KANE5 (20 m depth). Near surface DMS concentrations were below 0.4 nmol L$^{-1}$ at the station closest to the ice arch (KEN1) and were highest in association with the developing bloom, reaching 10 nmol L$^{-1}$ at KANE5. The maximum concentrations of DMS were mostly restricted to the upper 20 m of the water column, within or above the SCM when present.




### 3.5 North Water (NOW) transect

Variables measured across the NOW transect are presented in Figure 7. Sea surface temperatures were 1.0°C in the western part of the transect, between 3.5 and 4.0°C in the central part, and decreased to 2.3°C at the easternmost station (115). Sea surface salinity varied between 31.3 and 32.8 with highest values measured on the eastern edge of the transect, nearest to Greenland. Nitrate concentrations in surface water were below 0.04 µmol L$^{-1}$ across the whole transect, indicating post-bloom conditions similar to those found in the LS transect. Surface silicic acid concentrations varied between 0.3 and 6.5 µmol L$^{-1}$ with the lowest and highest values recorded at stations 111 and 101, respectively. Chl $a$ concentrations showed a subsurface peak at all stations with maximum values found at ca. 18 m depth at station 115 (4.7 µg L$^{-1}$), and at ca. 30 m depth at station 105 (4.3 µg L$^{-1}$). As observed further south at the mouth of Lancaster Sound, the phytoplankton assemblage was dominated by the centric diatom *Chaetoceros gelidus* and two unidentified *Chaetoceros* species at 3 stations of the transect (111, 108 and 101). Stations 115 and 105 were, for their part, numerically dominated by flagellates. Station 115 was also characterized by the presence of the prymnesiophyte *Phaeocystis pouchetii* (palmelloid stage) with an abundance reaching 503,700 cells L$^{-1}$, representing 9% of total cell counts at this station. Pennate diatoms were present across the transect, with concentrations increasing from west (16,425 cells L$^{-1}$ at station 101) to east (877,820 and 737,300 cells L$^{-1}$ at stations 111 and 115, respectively).

Concentrations of DMSP$_t$ were highest in the first 20-30 m of the water column, ranging from 34 to 88 nmol L$^{-1}$ at the near surface, with a distinct positive gradient from west to east. A subsurface peak was observed in the three most eastern stations (108, 111, and 115) with the highest concentrations of DMSP$_t$ (112 nmol L$^{-1}$, station 115) measured at 12 m depth. DMS concentrations in the near surface waters were relatively high and stable at 4.1-5.3 nmol L$^{-1}$ between stations 101 and 111 and reached 19.5 nmol L$^{-1}$ at station 115, the highest value measured during this campaign.

### 4 Discussion

During the joint ArcticNet/NETCARE cruise, summertime DMS distributions were studied in two regions of the High Canadian Arctic characterized by distinct ice edges: a first one featuring mainly ponded FYI, and a second one composed mainly of MYI. Both Barrow and Nares straits, as well as the contiguous regions of Lancaster Sound and North Water (northern Baffin Bay), embody significant oceanic gateways for Pacific-originating waters towards the North Atlantic (Jones et al. 2003). The results from the four transects conducted in these regions reveal distinctive features in DMS dynamics. The highlights of this study are discussed in the context of a predicted warmer Arctic, loss of perennial sea ice, and increase in the prevalence of seasonal FYI (Nghiem et al. 2007; Kwok and Rothrock 2009; Overland and Wang 2013; AMAP 2017).

### 4.1 Broad regional sea surface distributions of DMS

Over the entire study area, the distribution of sea surface concentrations of DMS (Fig. 8) ranged from 0.2 to 19.5 nmol L$^{-1}$, with a median of 4.4 nmol L$^{-1}$ and interquartile range of 2.5 nmol L$^{-1}$ (25th) and 8.2 nmol L$^{-1}$ (75th) (Table 3, n = 33). These values are similar to the measurements (range from 0.04 to 17.6 nmol L$^{-1}$ and a median of 2.47 nmol L$^{-1}$) conducted by Jarníková et al. (2018) at the same time of year (July-August) and in the same broad biogeographic sectors of the CAA and Baffin Bay. Altogether, these results show that previous measurements conducted in Baffin Bay and CAA earlier (April to June) or later (September-October) in the potentially productive season may not have been representative of summer conditions (see studies referenced in Table 3). These findings also bring further support to the hypothesis that local DMS





sources explain the mid-summer peaks of atmospheric MSA, a DMS proxy, in the High Arctic (Sharma et al. 2012; Becagli et al. 2019). Not surprisingly, considering our limited sea surface dataset (n = 33) and the overall complexity of the DMS cycle, no significant relationships were found between broad regional sea surface concentrations of DMS and biotic or abiotic variables. Global empirical relationships between DMS and other biogeochemical and physical variables are often

inconsistent and difficult to establish (Kettle et al. 1999). The Arctic, in particular, displays important patchiness in drivers of DMS dynamics (Galí and Simó 2010; Galindo et al. 2014; Jarníková et al. 2018), associated in great part with the presence of ice and its role in seeding under-ice algal blooms. The melt of sea ice and snow covers also influence surface water stratification and the ensuing shifts in salinity, temperature and solar radiation doses experienced by potential DMS-producing communities. The inherent heterogeneity that characterizes spatial distributions of DMS in the Arctic as well as

the presence of sea ice as a potentially critical driving force of these patterns warrants further investigations into underlying mechanisms.

### 4.2 The FYI edge in Barrow Strait and the adjacent Lancaster Sound

The seasonal sea ice zone (SIZ) in the Arctic is modulated by large interannual variability (Parkinson and Comiso 2013;

Simmonds 2015; Comiso et al. 2017; Serreze and Meier 2019). Correspondingly, the position of the ice edge in the Barrow Strait/Lancaster Sound area during spring may vary yearly from the mouth of the Sound on the east (80°W) to Lowther Island in Barrow Strait on the west (97°W) as revealed by the analysis of CIS ice charts by Peterson et al. (2008). On July 17, 2014, the ice edge was located approximately mid-way of this historical spatial range near the longitude of Prince Leopold Island (90°W, see Fig. 2A). Satellite imagery reveals that this distinct ice edge was already present a month prior to the arrival of

the icebreaker CCGS *Amundsen* in the area and that the eastern part of Lancaster Sound (east of 90°W) was already mostly ice-free by June 16, 2014 (data from CIS not shown). The ice cover in Barrow Strait, west of the ice edge, was composed mostly of FYI ca. 1 m thick covered with melt ponds at ca. 40% of its surface. On July 20, part of the ice diverged towards the east creating a small lead in the FYI near the northern tip of Somerset Island (Fig. 2A). The opportunity was taken to sample the western border of the lead, very close to the newly formed ice edge in order to capture the outflow of under-ice

waters. The predominantly eastward transport of water in the southern portion of the Strait is estimated at $14 \pm 4$ cm s$^{-1}$ annually and is strongest in late summer at $27 \pm 8$ cm s$^{-1}$ (Hamilton et al. 2013), suggesting that the residence time of seawater in the lead was short lived. Biogeochemical characteristics of the surface waters sampled on July 22-23 along the BS transect, particularly its southern area, thus likely reflect conditions prevailing in the ice-covered western portion of the Strait.

Vertical profiles from the BS transect (Fig. 4) in proximity to the newly formed ice edge indicate that an under-ice

phytoplankton bloom had developed in the ice-covered Barrow Strait area and was captured during our sampling as it exited the ice. This under-ice bloom coincided with relatively low salinities (ca. 31.5) and temperatures (ca. -1.5°C) within the surface waters. These results suggest that the bloom was linked to the development of a fresher water lens below the ice, likely resulting from the melting of snow and ice covers. Events that were also likely associated with an increase in light transmission through the ice, as observed by Galindo et al. (2014) for the 2010 and 2011 under-ice blooms in Allen Bay and

Resolute Passage. The relatively high nitrate concentrations measured in the upper 20 m (ca. 2.5 µmol L$^{-1}$) suggest that the bloom was still in its development phase. The maximum phytoplankton biomass measured in the lead and under the ice in adjacent stations (ca. 2 µg Chl *a* L$^{-1}$) was lower than the maximum value of ca. 10 µg L$^{-1}$ reported by Galindo et al. (2014) during under-ice blooms at the end of June 2011. The 5 times higher biomass measured during Galindo's study suggest that



turbulent mixing could bring additional nutrients to the under-ice bloom closer to shore. In stations of the BS transect however, a strong halocline had persisted at ca. 20 m, restricting the vertical input of nutrients to the upper part of the water column. Our results also indicate that the upwelling conditions which led to the formation of a massive under-ice bloom in Chukchi Sea (Arrigo et al. 2012; Cooper et al. 2016; Selz et al. 2018) were not present across the BS transect during our study. The dominance of the phytoplankton assemblage by pennate diatoms in the Barrow Strait region (BS transect station 305E as well as stations 304 and 305, see Fig. 1) suggests that the release of ice-associated algae (sympagic) likely contributed to the seeding of the under-ice bloom, as observed during similar under-ice blooms in the Barrow Strait region (Fortier et al. 2002; Galindo et al. 2014). These results agree with studies emphasizing the importance of ice algal communities as a seeding source during spring over oceanic regions when algal abundance in the water column is low (e.g., Arctic Ocean north of Svalbard by Kauko et al. (2018); Frobisher Bay in Davis Strait by Hsiao (1992)). The presence of species endemic to Arctic sea ice such as *Nitzschia frigida*, *Fragilariopsis cylindrus* and *Fragilariopsis oceanica* (Poulin et al. 2011) in the surface waters of the Barrow Strait region brings further support to the ice origin of this under-ice bloom.

The taxonomic composition of the drifting under-ice bloom at station 305E was also dominated by pennate diatoms, but with lower total cell abundance ($0.48 \times 10^6$ cells L$^{-1}$ at 305E) as compared to the two other Barrow Strait stations ($> 2.00 \times 10^6$ cells L$^{-1}$ at 304 and 305, data not shown), as well as slightly different species. The phytoplankton assemblage at 305E was similar to the one previously described by Galindo et al. (2014) for the under-ice bloom developing at a shallow station (50 m) in Allen Bay in 2011, located ca. 15 km west of 305E. In both studies, the under-ice bloom was dominated by pennate diatoms, with *Fossula arctica* and *Fragilariopsis oceanica* contributing 8.2 % and 7.8 %, respectively, to the total protist abundance at station 305E.

In the ice-free area of Lancaster Sound, the lower Chl *a* (0.2 to 1.2 µg L$^{-1}$) and nutrient concentrations measured in the 13-16 m depth SML as well as the presence of an SCM (Fig. 5) suggest that the bloom had reached a more advanced stage of development with part of the phytoplankton cells produced at the surface accumulating near the nitracline. Nitrate and silicic acid concentrations in the SML were limiting with values below 0.6 µmol L$^{-1}$ and 0.9 µmol L$^{-1}$, respectively, indicating that most of the nutrients, initially present in the surface, had already been consumed by the primary producers. The exhaustion of nitrate in surface waters, the deepening of the nitracline and the development of a SCM are typical features of summer conditions in several regions of the Arctic as demonstrated by a host of comprehensive investigations (Tremblay et al. 2008; Mundy et al. 2009; Martin et al. 2010; Ardyna et al. 2013; Brown et al. 2015; Steiner et al. 2015).

The striking difference between the phytoplankton assemblages in Lancaster Sound, dominated by centric diatoms, and Barrow Strait, dominated by pennate diatoms, suggests that the bloom in Lancaster Sound likely developed under ice-free conditions with no significant seeding from ice algae. This could indicate that part of the FYI pack broke and was flushed out of the eastern portion of the Sound before ice and snow conditions, as well as the potential presence of melt ponds, would have allowed the formation of an under-ice bloom. This interpretation is supported by MODIS images revealing that much of Lancaster Sound was ice-free a month prior to the ship's arrival (June 16, CIS data not shown). These results agree with findings from Hsiao (1992) showing a dominance of pennate diatoms in the ice algae and phytoplankton communities during spring in Frobisher Bay (Davis Strait), and a dominance of centric diatoms in open waters in summer post ice melt.

The Barrow Strait/Lancaster Sound FYI area was characterized by different vertical DMSP distributions in the two regions of the biogeographic sector under study, likely related to the distinctive phases of bloom development discussed above. In the newly formed lead of Barrow Strait, accumulation of DMSP$_t$ was highest in the surface waters with a peak of 115 nmol L$^{-1}$





at the northern edge of the lead (station 305E, depth of 2 m). This high value is in the same range as the under-ice water column concentrations of particulate DMSP ($DMSP_p$ of 99 and 185 nmol $L^{-1}$) observed by Galindo et al. (2014) in nearby Allen Bay. Despite seemingly varying vertical distribution patterns of *in vivo* fluorescence and $DMSP_t$ (see Fig. 4), the broad

fluctuations of these two biologically mediated variables displayed significant correlation ($r_s = 0.80$, $p < 0.001$, $n = 20$) suggesting that the bulk of $DMSP_t$ was intimately linked to algal biomass. In contrast, across much of the LS transect, particularly towards its southern portion, concentrations of $DMSP_t$ were highest near the nitracline, deeper in the water column (peak of 96 nmol $L^{-1}$ at 20 m, station 325). The role played by environmental drivers, such as nutrients, in the accumulation of DMSP-rich organisms at this depth was substantiated by the significant correlation found between water

column distributions of $NO_3^-$ and $DMSP_t$ ($r_s = -0.59$, $p < 0.001$, $n = 36$). However, contrary to patterns observed in the BS transect, concentrations of $DMSP_t$ bared no significantly association with *in vivo* fluorescence of chlorophyll in this part of the study area, suggesting that the bulk of algal biomass was not necessarily responsible for the variability in $DMSP_t$ concentrations in these waters characterized by mixed algal populations. The above results are not unexpected seeing as the nature of DMSP synthesis itself is highly species-specific (Keller et al. 1989) and subject to physiological up- or down-

regulation and excretion linked to environmental stressors (see review by Stefels et al. 2007). Assuming that almost all $DMSP_t$ was particulate (see Kiene and Slezak 2006), the $DMSP_t$:Chl *a* ratio can serve as an indicator for intracellular DMSP concentration. Averaged over the first 20 m of the water column, $DMSP_t$:Chl *a* ratios were 60 nmol µg$^{-1}$ at the four Barrow Strait stations and almost 3 times as high at the five Lancaster Sound stations (170 nmol µg$^{-1}$). This distinctive pattern could have resulted from differences in community composition or in physiological status of the algal communities, as well as their

bloom development phase. The numerical dominance of flagellates at certain stations of the LS transect (stations 322 and 323) and their presence in relatively high proportion at station 325 (second highest at 14%) suggest that the bloom was further along in its development than in the BS transect. This difference may help explain the large differences in Chl *a* normalized $DMSP_t$ between the two regions. However, in light of the scarcity in available information, the exact role played by community composition and cell physiology in shaping $DMSP_t$:Chl *a* ratios remains an open question.

Notwithstanding the lower $DMSP_t$:Chl *a* ratios in the BS transect, DMS levels were high in surface waters, ranging from 7.2 to 12 nmol $L^{-1}$ and revealed two hot spots at either end of the sampled transect (Fig. 4). One in association with a peak in $DMSP_t$ (115 nmol $L^{-1}$, 305E) and a second in conjunction with relatively low $DMSP_t$ (ca. 25 nmol $L^{-1}$, 305B) and Chl *a* (0.83 µg $L^{-1}$) at station 305B. Statistical analysis suggests that, in the waters exiting the FYI pack in Barrow Strait, variability in DMS concentrations was significantly associated with that of it's precursor $DMSP_t$ ($r_s = 0.76$, $p < 0.001$, $n = 20$) but was

most strongly associated with fluctuations in salinity. The highly significant negative correlation (Fig. 9) found between DMS and salinity ($r_s = -0.91$, $p < 0.001$, $n = 20$) in the upper ca. 80 m of the water column in this region suggests a strong physical control of DMS distributions associated with ice and snow melting processes. The generally sunny forecast in the days prior to the sampling excludes heavy rain as a significant contributor to this signal. During the thawing season, the increase in ice permeability and basal melting may trigger important releases of DMS in the waters just below the ice cover

(Trevena and Jones 2006; Kiene et al. 2007; Tison et al. 2010; Carnat et al. 2014). The formation of an upper fresher water "lens" associated with the FYI melt may also have led to a certain accumulation of DMS following its release from the sea ice. Furthermore, it cannot be totally excluded that the stratification of the upper water column ensuing from the melting ice could have entailed higher and longer exposures of phytoplankton communities to solar radiation with enhanced DMS production as a coping mechanism against light-induced stress via an antioxidant cascade (Sunda et al. 2002; Toole and





Siegel 2004; Vallina and Simó 2007; Galí and Simó 2010). Indirectly, DMS production could also have been stimulated through the possible increased availability of dissolved DMSP ($DMSP_d$) in the environment and its bacterially-mediated enzymatic conversion into DMS (Kiene et al. 2000). Laboratory salinity downshock experiments with batch cultures of diatoms and dinoflagellates have shown an increase in the excretion of cellular DMSP (Van Bergeijk et al. 2003) and an increase in the production of DMS (Stefels et al. 1996; Niki et al. 2007). A DMSP-related osmo-acclimation response to shifts in salinity (Stefels 2000) could be particularly beneficial for algae developing in highly fluctuating environments, such as in the Arctic during the thaw season, a phenomenon which could ultimately strengthen DMS production. The strength of the association between DMS and salinity in the waters however suggests that physical drivers exercised the greatest control over the distribution of DMS near the FYI ice edge.

Sea surface concentrations of DMS in the open waters of the Lancaster Sound transect ranged from 3 to 7 nmol L$^{-1}$ with no clear cross-channel pattern (Fig. 5). In this region, DMS concentrations peaked deeper in the water column (max of 12 nmol L$^{-1}$, station 300, 20 m depth) partly in association with the presence of DMSP-producing phytoplankton, as suggested by the significant positive correlation found between water column concentrations of DMS and $DMSP_t$ ($r_s = 0.45$, $p < 0.001$, n = 36). However, fluctuations in DMS throughout the water column were, once again, better correlated with those in salinity ($r_s = -0.72$, $p < 0.001$, n = 36) suggesting the continued importance of environmental drivers, such as salinity, in shaping DMS distributions in the later stages of bloom development in this part of the CAA during our study.

### 4.3 The MYI edge in Nares Strait and the adjacent North Water

Sea ice covers Nares Strait for most of the year (Münchow 2016). While the ice is typically landfast from December to June as a result of the formation of ice arches (ice bridges) at the northern entrance (around 83°N) and the southern exit (around 78°N), the ice is mobile from July to November (Kwok et al. 2010; Moore and McNeil 2018). As one of the major oceanic gateways of Arctic waters into the North Atlantic, the average depth-dependent flow through the Strait is always to the south (Münchow et al. 2007; McGeehan and Maslowski 2012) and drifting ice velocities through the channel are strongly correlated with local winds (Münchow 2016). The NS transect sampled during this study thus captured the flow of water exiting the northern ice pack towards Baffin Bay. In August 2014, the ice north of the head of Nares Strait, near Robeson Channel, was composed of MYI (Fig. 2C), a typical feature of this area (Kwok et al. 2005; Comiso 2012; Michel et al. 2015). A band of thick (> 1.2 m) FYI shaped a gradient between the northern MYI and southern open waters. While some FYI remained in the Strait, particularly southeast of the ice arch (along the Greenland coast), its concentration decreased from 7-8/10 to less than 3-4/10 and eventually to open waters west of Kane Basin.

Water column patterns of salinity along the NS transect were relatively uniform between stations with fresher waters reaching deeper into the water column at the most northern stations (Fig. 6). This pattern is consistent with the presence of Pacific-originating waters of lower salinity and density that enter the central Arctic Basin through Bering Strait and that partly flow south through Nares Strait as a sub-surface current (Jones et al. 2003). It may also reflect the southbound flow through Nares Strait of first-year or multiyear ice floes (Münchow 2016), or icebergs originating from the glaciers of Greenland or Ellesmere Island (Burgers et al. 2017) that can partially melt in transit and thus freshen the ocean surface waters the impact of which lessens to the south as the ice melts away. Vertical patterns of temperature along the NS transect showed well mixed waters down to 58 m in the station nearest to the ice edge (KEN1) and a progressive warming of the upper layers of the water column with decreasing latitude (Fig. 6). The rise in sea surface temperatures in the lower latitude stations of the NS transect





reflects the typical seasonality of heat exchange (summer heat uptake) between the open waters and the atmosphere (Maykut and McPhee 1995).

Reservoirs of nutrients throughout the water column at station KEN1, with 1.4 and 6 µmol L$^{-1}$ of nitrate and silicic acid, respectively, were at the lower end of expected pre-bloom values for Pacific-derived water of the same salinity in the higher Arctic (Tremblay et al. 2002). As the sampling stations progressed to the south, a drawdown of both those nutrients, associated with the development of phytoplankton biomass, was evident at the surface of the vertical profiles (Fig. 6). At station KANE3, nutrients exhibited a swell-like pattern, associated with an increase of nutrients throughout the water column.

The presence of a sill (Bourke et al. 1989) with a shallower bottom (depth of 230 m at 79°54'N) just north of KANE3 and the potential localized upwelling of nutrients could help explain this pattern.

Concentrations of Chl $a$ in the waters exiting the MYI pack were low (< 0.5 µg L$^{-1}$ at KEN1) throughout the water column while they rose with decreasing latitude in the rest of the NS transect. Chl $a$ concentrations reached a first surface peak of 2 µg L$^{-1}$ at KEN3 which then continued in subsurface waters. A SCM of 3 µg L$^{-1}$ was already present at KEN3 at ca. 24 m

depth and reached 8 µg L$^{-1}$ at ca. 35 m depth at KANE1. As this first bloom petered out, the upward supply of nitrate over a sill (230 m depth) located at latitude 79°54'N (Bourke et al. 1989) likely triggered another bloom in Kane Basin, with concentrations reaching 10 µg L$^{-1}$ at ca. 35 m depth at KANE5. Along this transect, variations in the phytoplankton biomass proxy, Chl $a$ $in$ $vivo$ fluorescence, were strongly associated with the availability of nitrate throughout the water column ($r_s$ = -0.92, p < 0.001, n = 44) reinforcing the role played by nutrients in shaping the vertical distribution of primary producers

as well as their control over nutrient drawdown.

In surface waters near the MYI edge, the phytoplankton community (dominated by unidentified flagellates and Prymnesiophyceae, KEN1, Table 2), showed a moderate abundance (1.3 × 10$^6$ cells L$^{-1}$, data not shown), suggesting that the initiation of a phytoplankton bloom had not yet occurred in waters underneath the northern ice pack. The presence of sufficient amounts of nutrients in the surface waters near the ice edge points towards light availability as the primary limiting

factor for the proliferation of primary producers under the ice. In seasonally ice-covered seas, the growth of shade-adapted algal cells may begin once a critical incident irradiance threshold is reached at the ice-water interface (Horner and Schrader 1982; Gosselin et al. 1986). These results are in sharp contrast to the patterns observed in the waters exiting the ponded FYI in Barrow Strait where a bloom had already begun to develop underneath the ice. The drawdown of silicic acid in the following NS transect stations concurred with the development and dominance of diatoms (see Table 2), notably centrics

such as $Chaetoceros$ spp. (5-20 µm) and $Chaetoceros$ $gelidus$, an assemblage similar to those previously described in the LS transect (as well as in the NOW transect later discussed). Species of the genus $Chaetoceros$ were thus widespread throughout the study area, as previously reported in the Canadian High Arctic (Booth et al. 2002; Ardyna et al. 2011; Poulin et al. 2011). In proximity to the northern ice edge in Nares Strait (KEN1), concentrations of DMSP$_t$ and DMS were rather modest throughout the water column (< 16 nmol L$^{-1}$ and < 0.4 nmol L$^{-1}$, respectively). These results reinforce the notion that

autotrophic and heterotrophic processes associated with the production of DMSP and DMS in the waters under the thick non-ponded MYI may have only truly taken off upon reaching ice-free, light-sufficient conditions found farther south. This is again in cutting contrast with DMSP and DMS patterns observed at the Barrow Strait ponded ice edge. Surface peaks of 27 nmol DMSP L$^{-1}$ and 2.6 nmol DMS L$^{-1}$ were measured in the following station (KEN3) adding support to the requisite of suitable doses of solar radiation to ensure the development of microalgae in ice-covered waters of the Arctic (Horner and

Schrader 1982; Gosselin et al. 1986) and the ensuing production of S compounds. In the three southernmost stations of the



Nares Strait transect, a subsurface maximum of $DMSP_t$ was present at ca. 20 m depth with a high value of 59 nmol $L^{-1}$ reached at KANE5 likely in association with an increase in autotrophic biomass fueled by nutrients near the sill, hitherto discussed. Maximal concentrations of DMS were, for the most part, confined to the upper 20 m of the water column within or above the SCM, with a high value of 10 nmol $L^{-1}$ reached at KANE5. Along this transect, variations in the vertical

distribution of DMS were significantly correlated with its precursor $DMSP_t$, however the strongest association was found between variations in DMS and seawater temperature ($r_s = 0.81$, $p < 0.001$, $n = 44$) likely reflecting seasonal warming of the ice-free surface waters and ensuing development of DMS-producing organisms. The significant positive correlation found between concentrations of DMS and *in vivo* fluorescence of chlorophyll ($r_s = 0.64$, $p < 0.001$, $n = 44$) throughout the water column in Nares Strait reinforces this suggestion (Fig. 9).

Ratios of $DMSP_t$:Chl *a* (ranging from 10 to 23 nmol $\mu g^{-1}$) averaged over the first 20 m of the water column of the NS transect were low compared to those found in the Lancaster Sound transect (max of 170 nmol $\mu g^{-1}$). Taking into account that our $DMSP_t$:Chl *a* ratios include both particulate and dissolved pools, and considering that dissolved DMSP typically contributes a small fraction of $DMSP_t$ (although highly variable; Kiene et al. 2000; Kiene and Slezak 2006), these values are nonetheless similar to previously reported $DMSP_p$:Chl *a* ratios with a maximum of 39 nmol $\mu g^{-1}$ (Luce et al. 2011) and a maximum of

17 nmol $\mu g^{-1}$ (Matrai and Vernet 1997), at diatom-dominated stations of the Canadian High Arctic and of the Barents Sea, respectively.

Along the ice-free west-east transect in the North Water (NOW), patterns of temperature and salinity (Fig. 7) revealed the interactions between the southward advection of fresh and cold Arctic waters along Ellesmere Island and saltier and warmer Atlantic waters flowing northward along western Greenland via the West Greenland Current (WGC) (Curry et al. 2011;

Münchow et al. 2015). Surface water concentrations of nitrate were below 0.04 $\mu mol$ $L^{-1}$ across the entire transect, exposing more mature blooming stage conditions similar to those found in the LS transect. As such, maximal accumulation of biomass occurred below the surface in most stations along the NOW transect in association with the nitracline (Spearman's rank correlation between *in vivo* fluorescence and $NO_3^-$, $r_s = -0.86$, $p < 0.001$, $n = 42$). The phytoplankton assemblage along the NOW transect was similar to the ones observed further south at the mouth of Lancaster Sound and further north along Nares

Strait. In the surface waters of stations 101, 108 and 111, the phytoplankton assemblage was dominated by the centric diatom *Chaetoceros gelidus* and two unidentified *Chaetoceros*, while flagellates numerically dominated the community at stations 105 and 115. The later station, located near the West Greenland Coast, was also characterized by the presence of palmelloid colonial cells of *Phaeocystis pouchetii*, reaching 9% of total cell count ($0.5 \times 10^6$ cells $L^{-1}$). Although the success and geographical range of *Phaeocystis* in the Northern Hemisphere are still poorly understood (Lovejoy et al. 2002; Schoemann

et al. 2005; Tremblay et al. 2012), particularly in the Canadian Arctic, the co-dominance of species of *Phaeocystis* has been shown to occur in the waters of the West Greenland Current (Fragoso et al. 2017) and Labrador fjords (Simo-Matchim et al. 2017).

*Phaeocystis* is widespread across the globe, including in high boreal and arctic waters (Verity et al. 2007) and its blooming has been linked to vast amounts of DMSP in the marine environment (van Duyl et al. 1998; Stefels et al. 2007; Asher et al.

2017). In this study, the presence of a DMSP hotspot (up to 113 nmol $L^{-1}$ at ca. 12 m depth) in the upper waters of the easternmost station 115 of the NOW transect may be partially explained by the occurrence of *Phaeocystis pouchetii* as well as the numerical dominance of unidentified flagellates, including potentially DMSP-rich species (Keller 1989). In the rest of the NOW transect, maximal concentrations of $DMSP_t$ in the first 30 m of the water column were lower, especially in the



westernmost station 101 (max of 35 nmol L$^{-1}$). This pattern likely reflects the signature of colder, fresher waters flowing south from Nares Strait along the western side of Baffin Bay with inferior concentrations of autotrophic biomass and DMSP$_t$. Ratios of DMSP$_t$:Chl $a$, averaged over the first 20 m of the water column at each station of the NOW transect, were very wide ranging with values fluctuating between 20 and 153 nmol µg$^{-1}$. The extent of these values is similar to the range of DMSP$_t$:Chl $a$ ratios (52–182 nmol µg$^{-1}$) found in the same sectors of the Arctic by Jarníková and collaborators (2018). Sea surface concentrations of DMS along the NOW transect were relatively stable at 4.1-5.3 nmol L$^{-1}$ between stations 101 and 111 and reached 19.5 nmol L$^{-1}$ at station 115. The occurrence of a localized DMS hotspot in the surface waters of the later station corresponded to a peak in DMSP$_t$ and the presence of *Phaeocystis* sp., a unicellular algal species known to be able to enzymatically convert DMSP into DMS and acrylic acid (Stefels and Dijkhuize 1996). The potential direct production of DMS by *Phaeocystis* sp. may have contributed to the heightened concentrations of DMS at this station. Variations in vertical profiles of DMS along the NOW transect were tightly coupled to those of DMSP$_t$ ($r_s = 0.85$, $p < 0.001$, $n = 42$). Because of the inherent complexity in the cycling of methylated sulfur compounds, involving biotic and abiotic factors of the environment, mismatches between DMS and algal biomass and DMSP are not uncommon, especially in temperate and subtropical waters (Archer et al. 2009; Dacey et al. 1998; Simó and Pedrós-Alió 1999; Vila-Costa et al. 2008). However, in seasonally light-limited polar waters, DMS tends to peak approximately simultaneously with phytoplankton biomass and with the concentration of its phytoplanktonic precursor, DMSP (Galí and Simó 2015). Although the strength of the association weakens when considering the entire study area (all stations within the 4 regional transects), vertical patterns of DMS and DMSP$_t$ remain significantly and positively correlated ($r_s = 0.64$, $p < 0.001$, $n = 142$) reinforcing the view that the dynamics of these two S compounds are broadly in phase during summer in the Arctic as observed elsewhere (Galí and Simó 2015).

**5 Synthesis**

**5.1 Increase in FYI at the expense of MYI: significance for DMS dynamics**

Results gathered through the NETCARE campaign in the Canadian Arctic Archipelago and Baffin Bay in July-August 2014 show distinct ocean DMS dynamics in relationship to two contrasting ice edges in terms of their age, developmental stage and the presence of melt ponds at their surface. Waters exiting the ponded FYI in Barrow Strait and sampled at the edge of a newly-formed lead in the region were characterized by a mixed phytoplankton community with pennate diatoms dominating the assemblage. Although indicators of biomass were of moderate magnitude (Chl $a < 2.1$ µg L$^{-1}$), concentrations of DMSP and DMS were high in the surface waters with maxima of 115 nmol L$^{-1}$ and 12 nmol L$^{-1}$, respectively, suggesting that a bloom had already started to develop under the melt pond-covered ice through the potential seeding of autotrophic organisms from the ice. The strong negative association found between salinity and DMS points towards ice itself as an important vector for sea surface DMS, contributing to its seeding at the ice-sea interface as observed elsewhere (Trevena and Jones 2006; Kiene et al. 2007; Tison et al. 2010). Haline-driven stratification of waters under the ice cover likely promoted the physical accumulation of DMS. Alternately, the surface stratification may have favored the biological production of DMS. The formation of a fresher water lens at the surface of the water could have led to the entrapment of algal cells and to an increase in solar radiation exposure with heightened DMS production as a defense strategy against light-associated oxidative stress (Sunda et al. 2002; Toole and Siegel 2004; Vallina and Simó 2007; Galí and Simó 2010). The fresher water lens may also have indirectly stimulated DMS production through the possible enhancement of DMSP$_d$ availability and its



bacterial conversion into DMS, following an osmotic-related excretion of cellular DMSP (Stefels 2000; Van Bergeijk et al. 2003; Niki et al. 2007). Although biological processes cannot be completely ruled out, the strength of the association between DMS and salinity near the FYI edge suggests that physical drivers most strongly shaped DMS dynamics in Barrow Strait.

In contrast to the FYI-dominated region described above, the waters exiting the MYI-dominated region of Nares Strait did not exhibit the same potential under-ice development of autotrophic organisms. The phytoplankton community in the surface waters of the station sampled nearest to the ice edge was dominated by flagellates and Chl *a* concentrations were comparatively low ($< 0.5$ µg L$^{-1}$), as were the concentrations of DMSP$_t$ ($< 16$ nmol L$^{-1}$) and DMS ($< 0.4$ nmol L$^{-1}$). The development of a phytoplankton bloom, and increase in both DMSP and DMS concentrations, occurred several km (ca.

100 km, Station KEN3) away from the ice edge highlighting the requirement for sufficient light to initiate the growth of primary producers. One of the distinguishing features between the two ice edges was the presence/absence of melt ponds at their surfaces. This factor likely played a major role in driving the availability of light through the ice as suggested by Nicolaus et al. (2012), leading to the earlier onset of a bloom (Fig. 10) and shaping the associated DMS cycling under the ice in the Barrow Strait region where melt ponds covered ca. 40% of the total surface. Findings from this study are of

particular significance in light of the suggestion that regions of the CAA (Fortier et al. 2002; Mundy et al. 2014), the Beaufort Sea (Mundy et al. 2014) and Baffin Bay (Oziel et al. 2019) may hold regular, yet under-documented, under-ice phytoplankton blooms. The occurrence of these blooms may be linked to the fact that the archipelago is characterized by narrow waterways where landfast ice tends to linger longer, allowing advanced stages of ice melt to be reached prior to break up, and where shallow waters act to enhance the supply of nutrients into surface waters fueling the potential growth of under-ice blooms

(Michel et al. 2006). Autotrophic biomass accumulations below the Chukchi Sea ice cover described by Arrigo et al. (2012) bring further support to the possible widespread importance of these blooms in waters of the Arctic. Furthermore, FYI has become the prevailing type of ice in the Arctic at the expense of swiftly declining MYI (Comiso et al. 2008). As such, and because FYI tends to have greater areal melt pond coverage than MYI due to a smoother topography (Polashenski et al. 2012), climate-driven changes in sea ice dynamics may lead to modifications in the timing and frequency of under-ice

blooms, their role in seeding ice-edge blooms in summer (Strass and Nöthig 1996) and the associated production of DMS (Galí and Simó 2010; Levasseur 2013). It is also worth noting that the highest sea surface DMS concentration measured during this expedition was associated with the presence of *Phaeocystis* (STN115, West Greenland current), a genus for which a few modelling studies point towards a poleward expansion in its geographical extent (Cameron-Smith et al. 2011; Menzo et al. 2018) associated with the increased intrusion of warm Atlantic water masses in the Arctic (Neukermans et al. 2018).

Altogether, these factors in conjunction with the projected increase in melt pond cover and their temporal span (Agarwal et al. 2011; Stroeve et al. 2014; Holland and Landrum 2015; Liu et al. 2015) and the direct role melt ponds may play in the production of DMS (Gourdal et al. 2018) suggests that there is a need to review the potential production and cycling of DMS in ice-covered areas of the Arctic during summer. As thinner, younger and more dynamic icescapes may prevail in the Arctic, earlier and more ubiquitous under ice blooms may lead to earlier pulses of DMS through leads, cracks and edges of the ice

with implications for climate forecasting.

Recent modelling studies predict an increase of DMS emissions in the Arctic, predominantly associated with sea ice retreat, and inducing a negative climate feedback through the influence of atmospheric DMS on cloud formation and radiative forcing (Kim et al. 2018; Mahmood et al. 2019). Most models however consider the ice-atmosphere interface to be inert. Possible diffusion of DMS through porous ice during spring (Gourdal et al. 2019), as well as potential DMS pulses venting to the



atmosphere via melt ponds (Gourdal et al. 2018) and through cracks and leads in thinner ice and at ice edges (Hayashida et al. 2017, this study) could lead to a strengthening of the DMS-related "polar-cooling"' predicted by models under future climate warming. According to a recent remote sensing study of pan-Arctic summertime emissions of DMS, however, the future response of cloud radiative forcing (warming or cooling) to increasing DMS emissions from ice-free waters remains uncertain (Galí et al. 2019). Modifications in anthropogenic emissions of sulfur and transport to the Arctic, changes in

shipping, industrialisation and oil-gas extraction in the Arctic, as well as the potential for longwave cloud forcing (warming) to offset shortwave cloud forcing (cooling) may all have impacts on the net radiation budget of the Arctic (Galí et al. 2019), highlighting the need to improve our understanding of plankton-climate feedbacks in the current context of rapid ecosystem transformation.

**Data availability**

CTD Data and Metadata are available on the Polar Data Catalog (PDC) at www.polardata.ca/pdcsearch/PDCSearchDOI.jsp?doi_id=12713. Metadata for other variables are available on the PDC at www.polardata.ca/pdcsearch/PDCSearchDOI.jsp? doi_id=12145. Data are available upon request by contacting the first author.


**Author contributions**

M. Lizotte was responsible for a large part of the sampling as well as the data analysis and processing. M. Levasseur and M. Lizotte wrote the initial version of the paper together. Several co-authors provided specific data included in the paper and all co-authors contributed to the final edition of the paper.


**Competing interests**

The authors declare that they have no conflict of interest.

**Acknowledgements**

The authors wish to thank the captain of the CCGS Amundsen, Alain Lacerte, as well as his officers and crew for their support during the oceanographic campaign. We also want to thank Jonathan Gagnon for nutrient analysis, and Sylvie Lessard for the taxonomic analysis. This study received financial support from NETCARE (funded under the NSERC Climate Change and Atmospheric Research program), ArcticNet (Network of Centres of Excellence of Canada) and the NSERC Discovery Grant Program and Northern Research Supplement Program (M. Levasseur, M. Gosselin and J.-É.

Tremblay). This paper is a contribution to the research programmes of NETCARE, ArcticNet, and Québec-Océan.

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





**Figures**


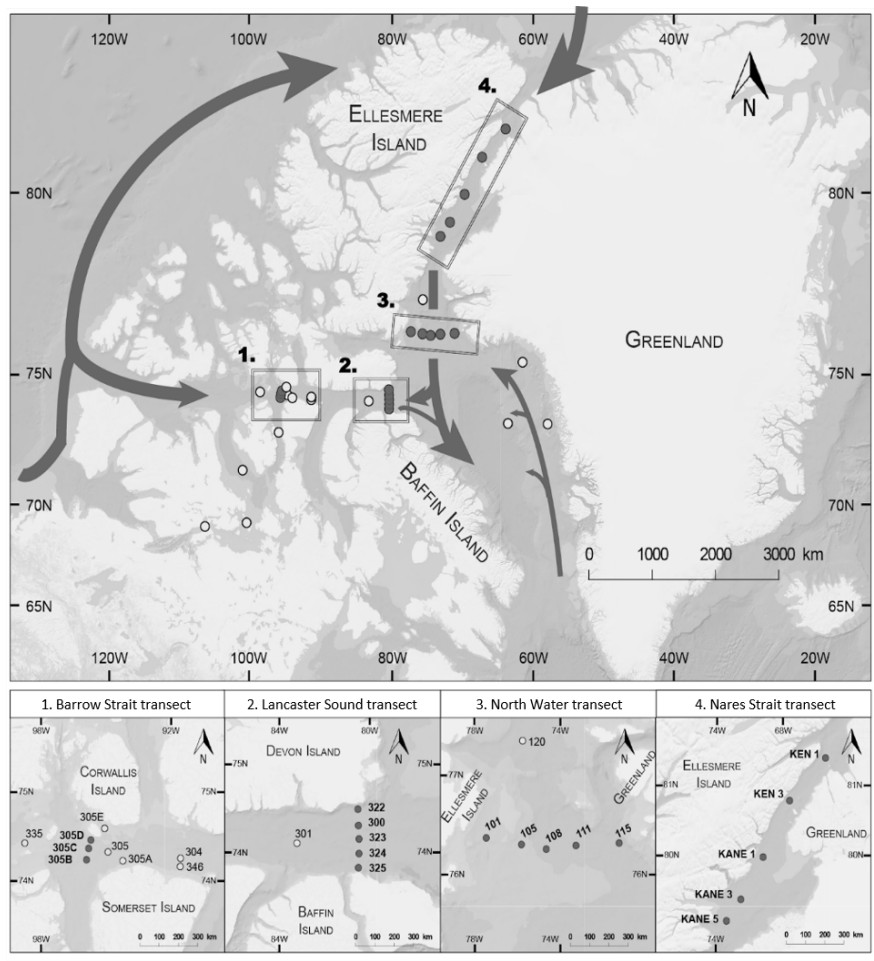

**Figure 1: Locations of the sampling stations in the eastern Canadian Arctic during the joint ArcticNet/NETCARE campaign in July-August 2014. Stations included in the four transects in Barrow Strait, Lancaster Sound, northern Baffin Bay (North Water) and Nares Strait, are represented by closed circles. Both closed and open circles are included in global observations of surface concentrations of DMS found in Figure 8. Major oceanic surface currents (in grey) are shown on the top panel.**





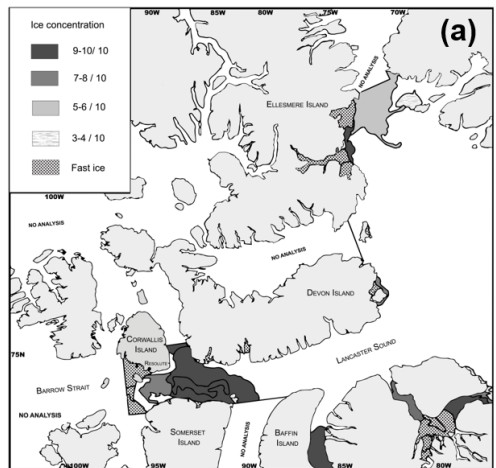

Figure 2: Ice charts adapted from the Canadian Ice Services (CIS) of Environment Canada showing the presence of ice edges in (a) Lancaster Sound with 9-10/10 ice concentrations (> 15 cm) extending from Devon Island into the Prince Regent Inlet between Somerset and Baffin islands (July 22, 2014); and in (b) Nares Strait with 9-10/10 ice concentrations near Petermann Glacier (August 1, 2014); Pannel (c) shows the presence of MYI (2 to 5+ years) at the entrance of Robeson Channel as well as a band of FYI (1) (Ease-Grid Sea Ice Age, Version 3 data set, Tschudi et al. (2016)). See the Canadian Ice Services Archives website for more details about sea ice characteristics. Note that Nares Strait includes Smith Sound, Kane Basin, Kennedy Channel, Hall Basin and Robeson Channel.

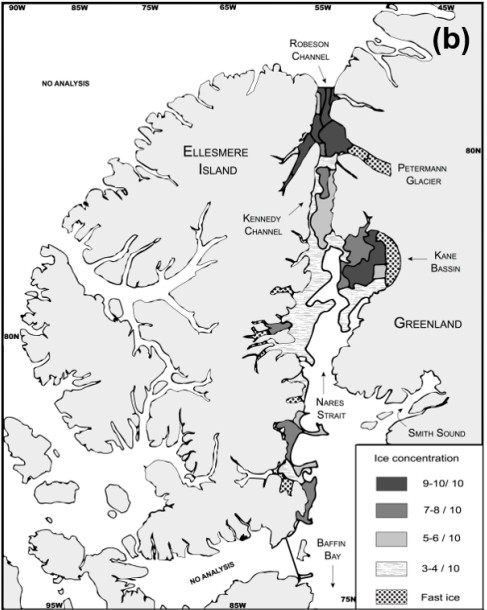

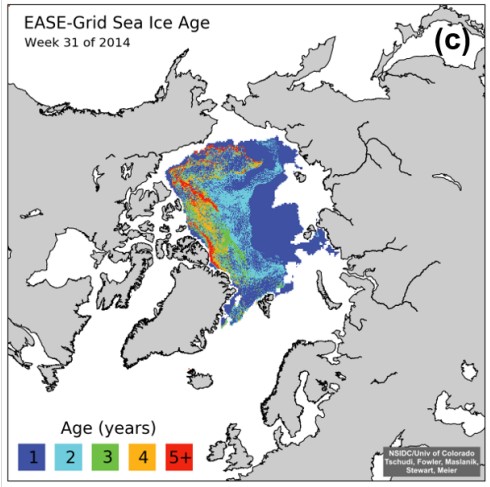




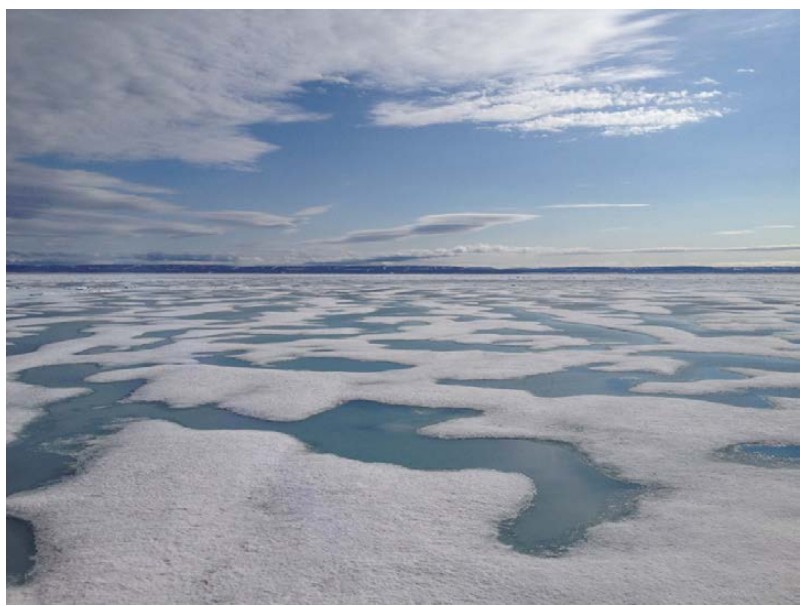

**Figure 3: Picture of ponded FYI at the western end of Lancaster Sound on July 20, 2014. Also see Gourdal et al. (2018) for more pictures of melt pond cover in the same region and same period.**



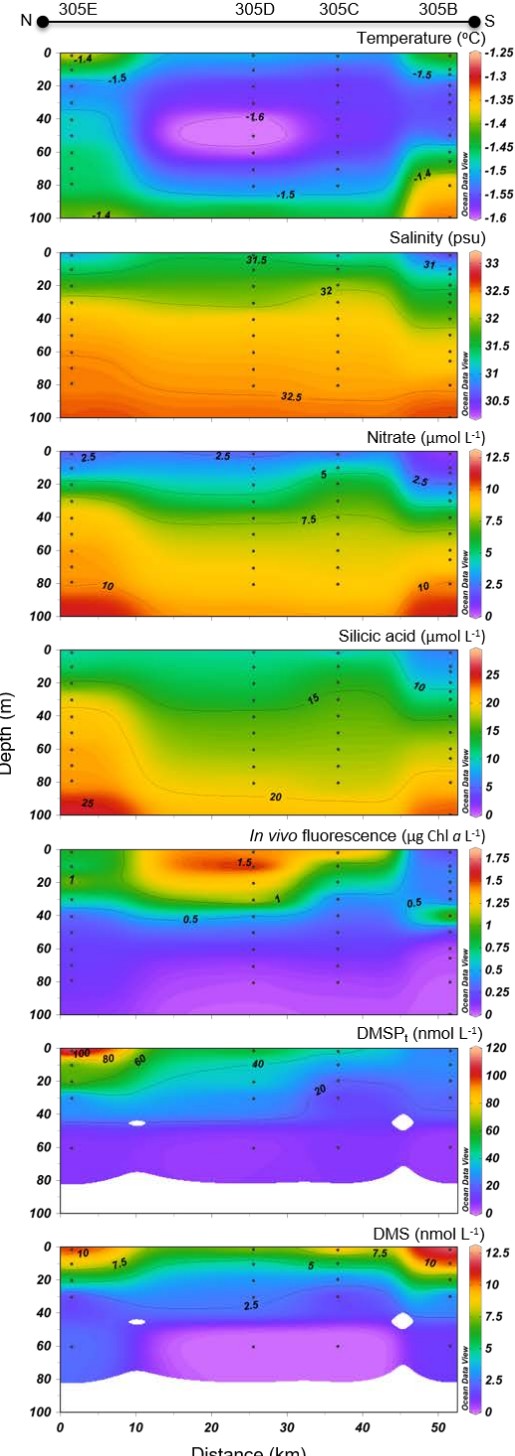

**Figure 4: Barrow Strait (BS) cross section of the vertical distributions of temperature, salinity, nitrate, silicic acid, *in vivo* fluorescence, DMSP$_t$ and DMS.**




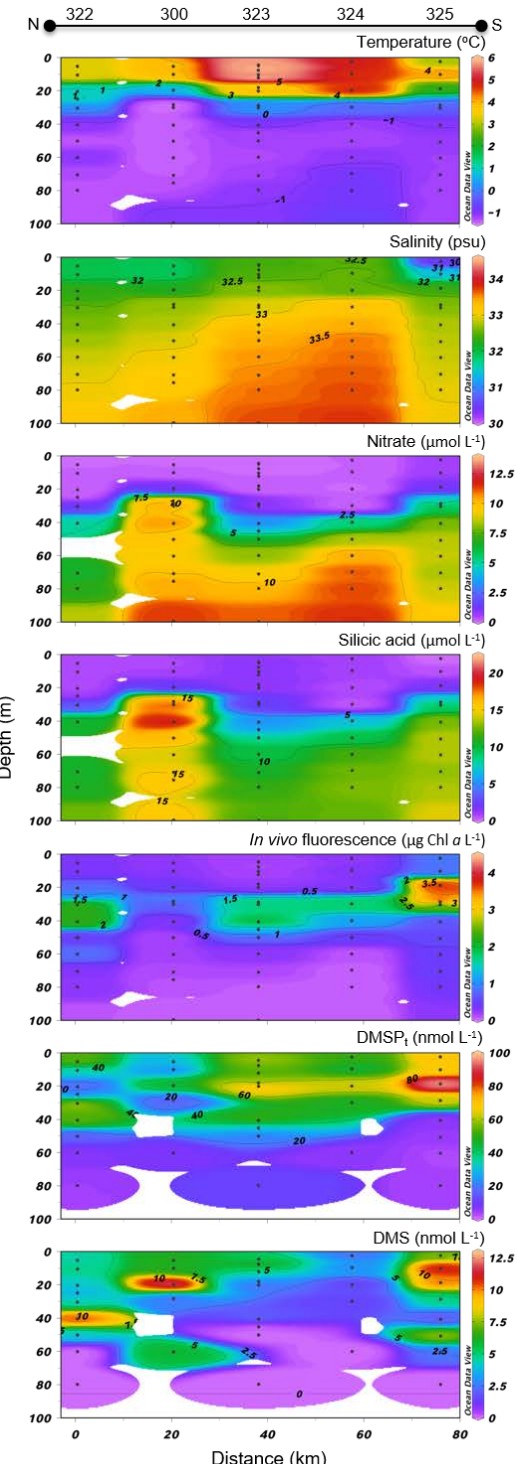

**Figure 5: Lancaster Sound (LS) cross section of the vertical distributions of temperature, salinity, nitrate, silicic acid, *in vivo* fluorescence, DMSP$_t$ and DMS.**



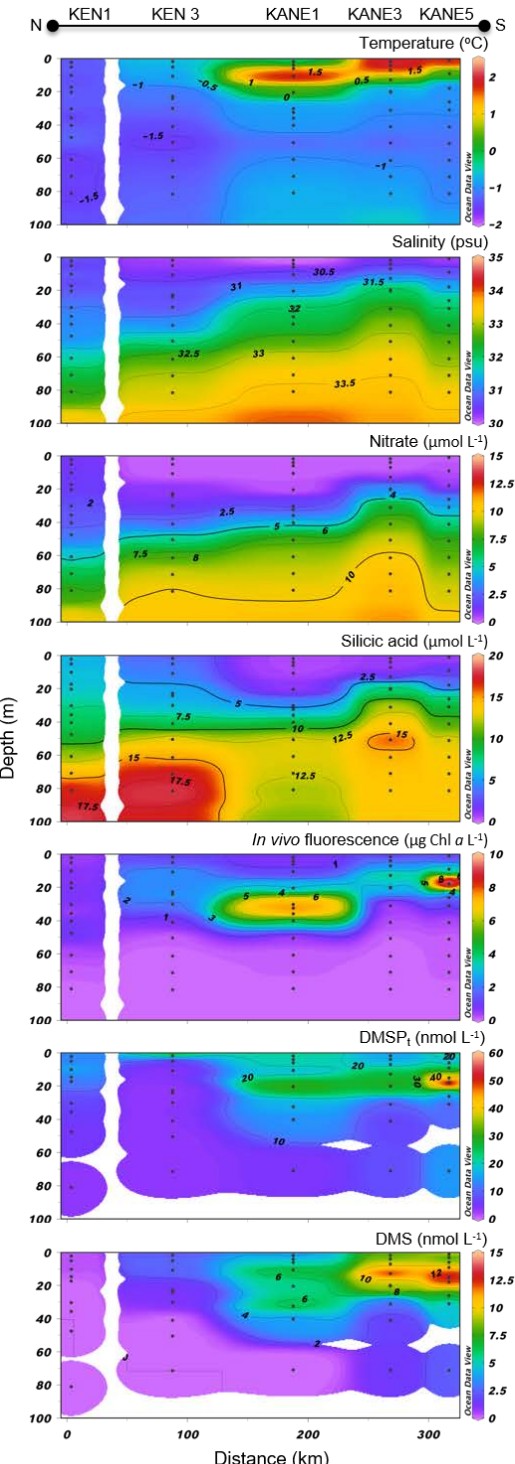

**Figure 6: Nares Strait (NS) cross section of the vertical distributions of temperature, salinity, nitrate, silicic acid, *in vivo* fluorescence, DMSP$_t$ and DMS.**



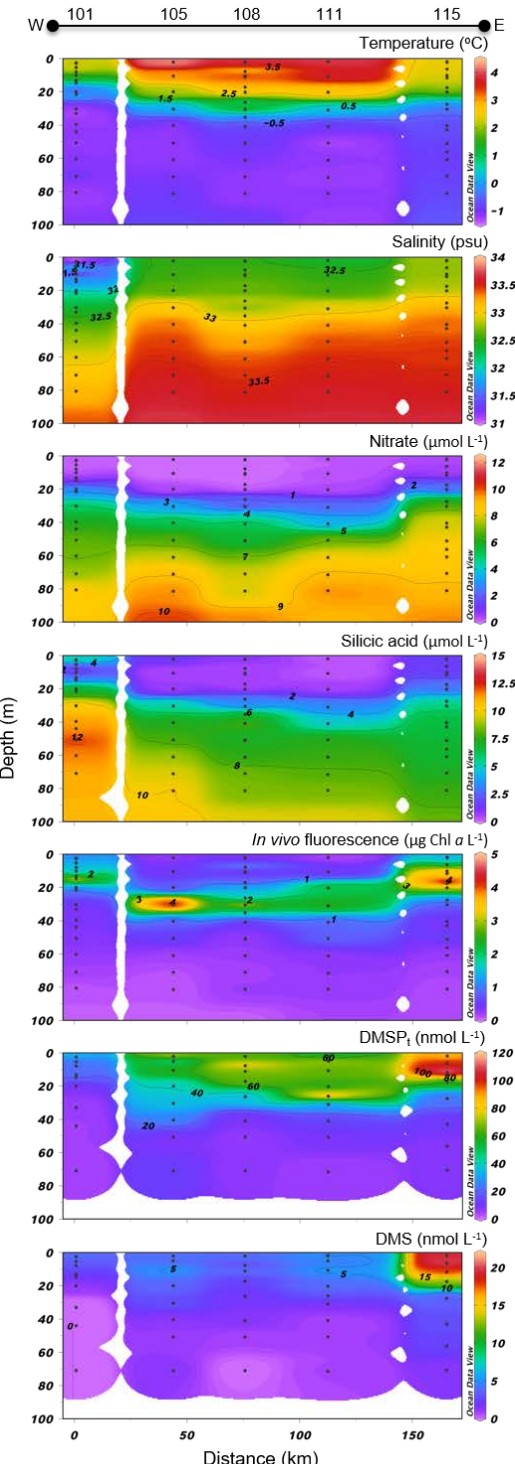

**Figure 7: North Water (NOW) cross section of the vertical distributions of temperature, salinity, nitrate, silicic acid,** *in vivo* **fluorescence, DMSP$_t$ and DMS.**



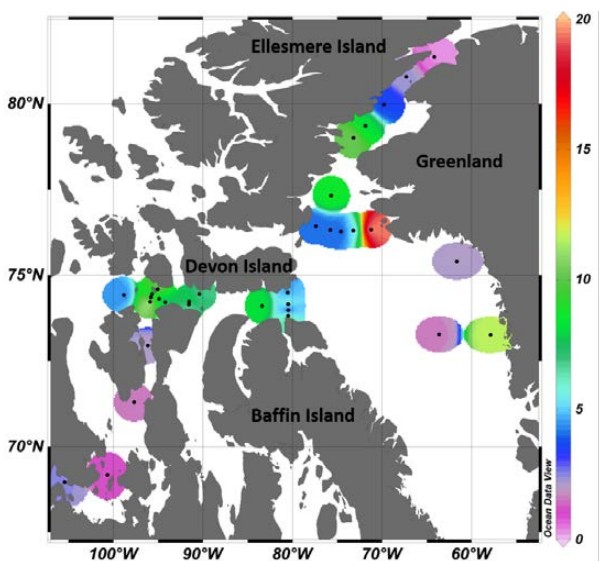


**Figure 8: Surface DMS concentrations (nmol L$^{-1}$) measured at 33 stations across the Canadian Arctic, during July-August 2014.**

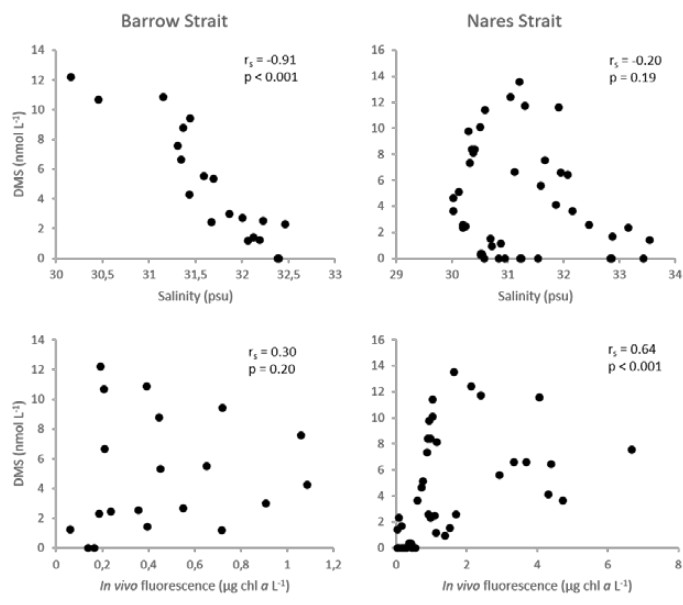


**Figure 9: Scatter diagrams of DMS, as a function of salinity and *in vivo* fluorescence, in the upper ca. 100 m of the water column in the Barrow Strait transect (n = 20) on the left and the Nares Strait transect (n = 44) on the right. r$_s$: Spearman's rank correlation.**



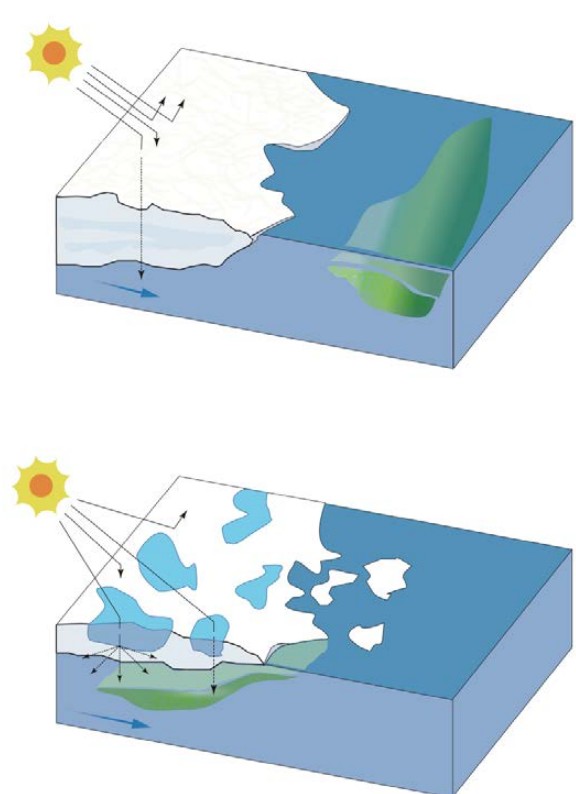

**Figure 10: Conspicuous alterations in the Arctic Ocean are underway and include reductions in snow cover, sea ice extent and thickness, and increase in melt pond areal coverage, the occurrence of which is linked to profound modifications in light availability in surface waters below the ice and at its margin. How these physical changes will impact the dynamics of bloom-forming microorganisms and their production of the biogenic climate-active gas DMS are still unknown. The conceptual diagram depicts two types of ice edges (top panel MYI and lower panel FYI) and their potential role in modulation light penetration under**
**the ice pack and the development of phytoplankton blooms and associated DMS dynamics.**



**Tables**


**Table 1: Physicochemical characteristics of the surface waters sampled during the ArcticNet/NETCARE campaign during July-August 2014 grouped within different regions of the Arctic. The surface mixed layer depth is noted at MLD, nitrate ($NO_3^-$), silicic acid ($Si(OH)_4$) and phosphate ($PO_4^{3-}$). Ice-covered stations are marked with an asterix\*. Stations in bold are included in the vertical cross section transect figures (Figs. 4 to 7). Values that were not available are noted as 'n.a.'**


| Region | Station | Latitude | Longitude | Date (yyyy-mm-dd) | Sampling depth (m) | MLD (m) | Temp (°C) | Salinity (PSU) | $NO_3^-$ ($\mu$mol L$^{-1}$) | $Si(OH)_4$ ($\mu$mol L$^{-1}$) | $PO_4^{3-}$ ($\mu$mol L$^{-1}$) |
|---|---|---|---|---|---|---|---|---|---|---|---|
| Barrow Strait | 305a | 74° 12' 59.4" N | 94° 12' 54" W | 2014-07-22 | 2.0 | n.a. | -0.70 | 30.41 | 0.13 | 3.45 | 0.69 |
| | **305b** | 74° 13' 44.04" N | 95° 54' 28.08" W | 2014-07-23 | 1.6 | n.a. | -1.41 | 30.50 | 0.64 | 5.96 | 0.77 |
| | **305c** | 74° 21' 34.56" N | 95° 48' 36.36" W | 2014-07-23 | 1.9 | n.a. | -1.51 | 31.18 | 2.18 | 9.53 | 0.87 |
| | **305d** | 74° 27' 22.68" N | 95° 42' 10.08" W | 2014-07-23 | 1.5 | n.a. | -1.50 | 31.46 | 2.08 | 10.57 | 0.85 |
| | **305e** | 74° 35' 19.32" N | 95° 3' 42.84" W | 2014-07-23 | 1.6 | n.a. | -1.37 | 31.04 | 2.10 | 10.47 | 0.86 |
| | 305* | 74° 19' 6.24" N | 94° 54' 23.04" W | 2014-07-22 | 1.8 | 15 | -0.99 | 30.60 | 0.14 | 4.29 | 0.81 |
| | 304* | 74° 14' 45.996" N | 91° 31' 4.008" W | 2014-07-20 | 1.8 | 23 | -1.39 | 29.52 | 0.01 | 3.75 | 0.71 |
| | 346* | 74° 8' 57.984" N | 91° 31' 55.992" W | 2014-07-20 | 2.0 | n.a. | -1.52 | 30.60 | 0.02 | 4.53 | 0.73 |
| Lancaster Sound | **322** | 74° 29' 53.016" N | 80° 31' 30" W | 2014-07-18 | 5.3 | 13 | 3.31 | 31.96 | 0.11 | 0.76 | 0.35 |
| | **300** | 74° 18' 42.084" N | 80° 29' 53.016" W | 2014-07-18 | 5.3 | n.a. | 3.87 | 31.88 | 0.12 | 0.78 | 0.44 |
| | **323** | 74° 9' 21.996" N | 80° 28' 26.004" W | 2014-07-17 | 4.5 | 16 | 5.68 | 32.35 | 0.13 | 1.31 | 0.27 |
| | **324** | 73° 59' 6" N | 80° 28' 26.976" W | 2014-07-18 | 2.6 | n.a. | 4.83 | 32.51 | 0.13 | 0.86 | 0.23 |
| | **325** | 73° 49' 0.984" N | 80° 29' 52.98" W | 2014-07-19 | 2.4 | 14 | 2.99 | 30.46 | 0.58 | 0.08 | 0.48 |
| | 301 | 74° 6' 8.352" N | 83° 22' 36.372" W | 2014-07-19 | 2.2 | 15 | 2.84 | 31.16 | 0.02 | 0.79 | 0.46 |
| North Water | **101** | 76° 22' 59.988" N | 77° 24' 0" W | 2014-08-01 | 2.5 | 6,0 | 2.44 | 30.39 | 0.15 | 0.85 | 0.31 |
| | **105** | 76° 19' 3" N | 75° 45' 32.004" W | 2014-08-01 | 2.1 | 44 | 4.19 | 32.51 | 0.03 | 1.33 | 0.30 |
| | **108** | 76° 16' 13.008" N | 74° 35' 60" W | 2014-07-31 | 2.1 | 11 | 3.69 | 32.52 | 0.04 | 0.96 | 0.27 |
| | **111** | 76° 18' 24.012" N | 73° 12' 54" W | 2014-07-31 | 2.0 | 33 | 3.56 | 32.39 | 0.26 | 0.25 | 0.25 |
| | **115** | 76° 19' 54.984" N | 71° 11' 57.012" W | 2014-07-30 | 1.9 | 25 | 2.40 | 32.78 | 0.36 | 1.50 | 0.25 |
| Nares Strait | **KEN1** | 81° 21' 36.252" N | 63° 57' 21.672" W | 2014-08-03 | 2.1 | 58 | -1.33 | 30.73 | 1.4 | 6.02 | 0.53 |
| | **KEN3** | 80° 47' 43.728" N | 67° 18' 4.032" W | 2014-08-04 | 1.7 | 18 | -0.66 | 30.21 | 0.22 | 3.34 | 0.38 |
| | **KANE1** | 79° 59' 35.016" N | 69° 46' 38.172" W | 2014-08-04 | 1.8 | 10 | -0.26 | 29.97 | 0.20 | 0.66 | 0.34 |
| | **KANE3** | 79° 20' 46.032" N | 71° 51' 28.152" W | 2014-08-05 | 1.7 | 17 | 1.87 | 30.46 | 0.17 | 0.99 | 0.28 |
| | **KANE5** | 79° 0' 5.508" N | 73° 12' 16.452" W | 2014-08-06 | 1.7 | 13 | 1.49 | 30.29 | 0.26 | 0.96 | 0.31 |





**Table 2: Biogeochemical characteristics of the surface waters sampled during the ArcticNet/NETCARE campaign during July-August 2014 grouped within different regions of the Arctic. Values in italic between parentheses represent *in vivo* fluorescence. Ice-covered stations are marked with an asterix\*. Stations in bold are included in the vertical cross section transect figures (Figs. 4 to 7). Values that were not available are noted as 'n.a.'**

| Region | Station | Latitude | Longitude | Date (yyyy-mm-dd) | Sampling depth (m) | Chl *a* (µg L$^{-1}$) | DMSP$_t$ (nmol L$^{-1}$) | DMS (nmol L$^{-1}$) | Dominant phytoplankton taxa (% of total cells) |
|---|---|---|---|---|---|---|---|---|---|
| Barrow Strait | 305a | 74° 12' 59.4" N | 94° 12' 54" W | 2014-07-22 | 2,0 | 1.14 *(0.99)* | 44.9 | 7.2 | |
| | **305b** | 74° 13' 44.04" N | 95° 54' 28.08" W | 2014-07-23 | 1.6 | 0.19 *(0.19)* | 25.2 | 12.2 | |
| | **305c** | 74° 21' 34.56" N | 95° 48' 36.36" W | 2014-07-23 | 1.9 | 0.83 *(0.72)* | 42.8 | 9.4 | |
| | **305d** | 74° 27' 22.68" N | 95° 42' 10.08" W | 2014-07-23 | 1.5 | 1.05 *(1.06)* | 56.1 | 7.6 | |
| | **305e** | 74° 35' 19.32" N | 95° 3' 42.84" W | 2014-07-23 | 1.6 | 0.83 *(0.39)* | 114.7 | 10.9 | Pennate diatoms (29%) |
| | 305* | 74° 19' 6.24" N | 94° 54' 23.04" W | 2014-07-22 | 1.8 | 0.90 *(0.62)* | 45.6 | 7.6 | Pennate diatoms (65%) |
| | 304* | 74° 14' 45.996" N | 91° 31' 4.008" W | 2014-07-20 | 1.8 | 2.29 *(0.27)* | 71.6 | 8.7 | Pennate diatoms (69%) |
| | 346* | 74° 8' 57.984" N | 91° 31' 55.992" W | 2014-07-20 | 2,0 | n.a. *(0.94)* | 63.5 | 5.8 | |
| Lancaster Sound | **322** | 74° 29' 53.016" N | 80° 31' 30" W | 2014-07-18 | 5.3 | 0.23 *(0.26)* | 54.6 | 4.6 | Flagellates (24%) |
| | **300** | 74° 18' 42.084" N | 80° 29' 53.016" W | 2014-07-18 | 5.3 | n.a. *(0.14)* | 29.8 | 6.3 | |
| | **323** | 74° 9' 21.996" N | 80° 28' 26.004" W | 2014-07-17 | 4.5 | 0.14 *(0.10)* | 59.6 | 5.3 | Flagellates (37%) |
| | **324** | 73° 59' 6" N | 80° 28' 26.976" W | 2014-07-18 | 2.6 | n.a. *(0.19)* | 50.4 | 2.5 | |
| | **325** | 73° 49' 0.984" N | 80° 29' 52.98" W | 2014-07-19 | 2.4 | 0.64 *(0.59)* | 70.6 | 7.2 | Centric diatoms (37%) |
| | 301 | 74° 6' 8.352" N | 83° 22' 36.372" W | 2014-07-19 | 2.2 | 1.16 *(0.26)* | 53.1 | 8.2 | Centric diatoms (41%) |
| North Water | **101** | 76° 22' 59.988" N | 77° 24' 0" W | 2014-08-01 | 2.5 | 1.21 *(0.40)* | 34.5 | 4.1 | Centric diatoms (41%) |
| | **105** | 76° 19' 3" N | 75° 45' 32.004" W | 2014-08-01 | 2.1 | 0.29 *(0.22)* | 70.5 | 4.2 | Flagellates (31%) |
| | **108** | 76° 16' 13.008" N | 74° 35' 60" W | 2014-07-31 | 2.1 | 0.48 *(0.14)* | 64.2 | 3.9 | Centric diatoms (43%) |
| | **111** | 76° 18' 24.012" N | 73° 12' 54" W | 2014-07-31 | 2,0 | 0.30 *(0.21)* | 54.8 | 5.3 | Centric diatoms (33%) |
| | **115** | 76° 19' 54.984" N | 71° 11' 57.012" W | 2014-07-30 | 1.9 | 1.69 *(1.14)* | 88.1 | 19.5 | Flagellates (21%) |
| Nares Strait | **KEN1** | 81° 21' 36.252" N | 63° 57' 21.672" W | 2014-08-03 | 2.1 | 0.32 *(0.38)* | 13.5 | 0.37 | Flagellates (28%) |
| | **KEN3** | 80° 47' 43.728" N | 67° 18' 4.032" W | 2014-08-04 | 1.7 | 1.90 *(0.97)* | 26.8 | 2.33 | Centric diatoms (65%) |
| | **KANE1** | 79° 59' 35.016" N | 69° 46' 38.172" W | 2014-08-04 | 1.8 | 1.57 *(0.61)* | 22.4 | 3.61 | Centric diatoms (73%) |
| | **KANE3** | 79° 20' 46.032" N | 71° 51' 28.152" W | 2014-08-05 | 1.7 | 0.76 *(1.15)* | 16.0 | 8.12 | Centric diatoms (67%) |
| | **KANE5** | 79° 0' 5.508" N | 73° 12' 16.452" W | 2014-08-06 | 1.7 | 0.84 *(0.93)* | 23.8 | 0.78 | Centric diatoms (72%) |





 **Table 3: Number of observations (Obs.), mean, median, minimum, maximum, standard deviation (SD) and interquartile range (IQR, 25th and 75th percentiles) of surface DMS concentrations from the Canadian Arctic Archipelago (CAA) and Baffin Bay biogeographic sectors from spring to autumn. DMS observations from this study include the transect stations (BS, LS, NS, and NOW) as well as other stations in the CAA and Baffin Bay sampled during the ArcticNet/NETCARE campaign (see Figure 8 for DMS spatial distribution map).**

| Biogeographic sector | Date | | Obs. | Mean | Median | Minimum | Maximum | SD | IQR 25th-75th | Study |
|---|---|---|---|---|---|---|---|---|---|---|
| | Month | Year | n | | | (nmol L$^{-1}$) | | | | |
| Baffin Bay | April | 1998 | 56 | 0.17 | 0.13 | nd | 0.72 | 0.15 | | Bouillon et al. 2002 |
| | May | 1998 | 53 | 0.65 | 0.33 | 0.07 | 6.74 | 1.02 | | |
| | June | 1998 | 55 | 1.08 | 0.70 | 0.04 | 4.59 | 1.11 | | |
| Baffin Bay and CAA | Oct/Nov | 2007 | 20 | | | 0.05 | 0.08 | | | Luce et al. 2011 |
| Baffin Bay and CAA | September | 2008 | 15 | 1.27 | 0.80 | 0.50 | 4.80 | 1.09 | | Motard-Côté et al. 2012 |
| Baffin Bay and CAA | July/August | 2015 | 165 | 3.29 | 2.47 | 0.04 | 17.58 | 2.72 | 1.62-3.89 | Jarníková et al. 2018 |
| Baffin Bay and CAA | July/August | 2014 | 33 | 5.39 | 4.35 | 0.23 | 19.53 | 3.87 | 2.39-7.91 | This study |
