# Peer review of "Phytoplankton and dimethylsulfide dynamics at two contrasting Arctic ice edges"

_Biogeosciences, 2019_

## Referee Comment (RC1) · Anonymous Referee #1 · 7 Jan 2020

"Phytoplankton and dimethylsulfide dynamics at two contrasting Arctic ice edges" by M. Lissotte et al. was reviewed.

In this paper, the authors focused on the relationship between phytoplankton and DMS dynamics at the different type of icescapes, i.e. ice edges dominated by first-year and multi-year ices. The authors well documented the different characteristics for DMS production between the icescapes. In general, this paper is suitably written and I totally agree with the authors' discussion. I recommend that this paper will be published in this journal after small correction of technical issues.

Specific comments Page 4, Line 115: "Sound" → "sound". The same errors can be found. Please correct it.

[Figure]

Page 4, Line 148: The term "chl a" is firstly mentioned here. The authors mentioned the abbreviation at Page 5 Line 155, but it should be shown here. Also, the abbreviations are shown as both "chl a" and "Chl a". Please unify the abbreviation.

Page 5, Line 156: "onto 25-mm filters" → "onto a 25-mm filter"

Page 5, Line 156: Phytoplankton pigments were extracted... → Phytoplankton pigments on the filter were extracted...

Page 5, Lines 168-170: This sentence should be reconsidered. The authors may merge the measurement steps of both natural sample and standard in error.

GC column: Please indicate the type of GC column of two GC systems.

Page 6, Line 189: What is the "proprietary trap" here? Please explain the detail.

Figure 2: The size of letters on the map are too small. Please resize it.

Figure 10: For FYI diagram, the relationship between phytoplankton bloom and light availability is clearly indicated, but I'm afraid that the reader may not catch what the authors would like to show in MYI diagram. Please modify the MYI diagram to show the relationship of phytoplankton abundance and light availability. Also, the second sentence (How these physical changes...) may be omitted from the figure caption.

---

## Referee Comment (RC2) · Alison Webb (Referee) · 21 Jan 2020

This manuscript is a well written and comprehensive review of different parameters determining the dynamics of DMS and DMSP along the ice edges around Greenland and the Canadian arctic. It is a good addition to the currently limited dataset on the relationship between DMS and the sea ice, and highlights the differences between single and multi-year ice. As the Arctic experiences greater ice loss, the importance of the single year ice is going to become more important to the global climate cycle of DMS, but we need to understand the difference to multi-year ice too. This particular aspect was well covered in the final discussion. I recommend this manuscript for publication with the following minor revisions.

[Figure]

**BGD**

L71. Please include a comment to the potential of Bacterioplankton to produce DMSP (Curson et al 2017) as well as break it down.

L426. 'Bared no significantly' should be 'bore no significant'

L577 Reference spell check Stefels and Dijkhuizen 1996

Fig 2 caption. Spell check Pannel to panel

Table 1 and 2 captions. Both these captions are the same, despite different parameters highlighted in the tables. Can the captions be made more unique to describe the data in each table?

---

## Author Comment (AC1) · 4 Feb 2020

Anonymous Referee #1 "Phytoplankton and dimethylsulfide dynamics at two contrasting Arctic ice edges" by M. Lizotte et al. was reviewed.

In this paper, the authors focused on the relationship between phytoplankton and DMS dynamics at the different type of icescapes, i.e. ice edges dominated by first-year and multi-year ices. The authors well documented the different characteristics for DMS production between the icescapes. In general, this paper is suitably written and I totally agree with the authors' discussion. I recommend that this paper will be published in this journal after small correction of technical issues. Author's response. We thank anony-

mous referee #1 for their constructive review of the manuscript. Below we address each point brought up by referee #1.

Note from the authors. Because a phrase was added on L70 following the comments of referee #2, the numbering of the lines has changed. We take into account this new numbering in our response to both referees.

Specific comments Page 4, Line 115: "Sound" → "sound". The same errors can be found. Please correct it. Author's response. Yes, the word "sound" on its own should not be capitalized. Author's changes in manuscript. A search for "Sound, as a stand-alone word (not part of a name such as Lancaster Sound) was made throughout the text and replaced with "sound". L115 (now L118) "Sound" was changed to "sound" L362 "Sound" was changed to "sound" L418 "Sound" was changed to "sound"

Page 4, Line 148: The term "chl a" is firstly mentioned here. The authors mentioned the abbreviation at Page 5 Line 155, but it should be shown here. Also, the abbreviations are shown as both "chl a" and "Chl a". Please unify the abbreviation. Author's response. We agree with the referee: on L148 (now L151), chlorophyll a should be written out and there were different forms of the abbreviation of chlorophyll a. Author's changes in manuscript. On L148 (now L151), the word "chlorophyll a" was added and we changed "chl a" for "(Chl a)". On L158, we changed "chlorophyll a (chl a)" to simply "Chl a", Throughout the text we kept the capitalized form Chl a, just as can be found in other papers published in Biogeosciences, e.g. "Dutkiewicz, S., Hickman, A. E., and Jahn, O.: Modelling ocean-colour-derived chlorophyll a, Biogeosciences, 15, 613–630, https://doi.org/10.5194/bg-15-613-2018, 2018."

Page 5, Line 156: "onto 25-mm filters" → "onto a 25-mm filter" Author's response. We agree to the modification. Author's changes in manuscript. On L156 (now L158-L159), the words "onto 25-mm filters" were changed to "onto a 25-mm filter".

Page 5, Line 156: Phytoplankton pigments were extracted. . . → Phytoplankton pigments on the filter were extracted. . . Author's response. We agree with the

proposed modification. Author's changes in manuscript. On L156 (now L159), we modified "Phytoplankton pigments were extracted..." by "Phytoplankton pigments on the filter were extracted..."

Page 5, Lines 168-170: This sentence should be reconsidered. The authors may merge the measurement steps of both natural sample and standard in error. GC column: Please indicate the type of GC column of two GC systems. Author's response. We agree that precision about GC columns can be added. Author's changes in manuscript. Information pertaining to the type of GC columns used was added to the text.

L173-174: Gaseous samples were then analyzed using a Varian 3800 gas chromatograph (GC), equipped with a Pulsed Flame Photometric Detector (PFPD) and a capillary column (DB-5ms, 60m x 320um x 1um) L195-L196: Once separated by the GC capillary column (DB-5ms, 30m x 250um x 0.25um), volatile compounds were ionized and directed to the mass selective quadrupole of the MS.

Page 6, Line 189: What is the "proprietary trap" here? Please explain the detail. Author's response. The trap mentioned on L189 (now L192) is "proprietary" meaning that Teledyne Tekmar has the proprietary rights to its composition. We suggest taking this word out to avoid confusion and add extra details instead. Author's changes in manuscript. On L189 (now 192), the words "u-shaped proprietary trap" were changed to "u-shaped trap for volatile organic compounds (Teledyne Tekmar Stamp 9 Trap)."

Figure 2: The size of letters on the map are too small. Please resize it. Author's response. Yes, we agree with both the comment and the suggestion. Author's changes in manuscript. The size of the letters on the map were made larger. A new version of Figure 2 was added to the manuscript.

Figure 10: For FYI diagram, the relationship between phytoplankton bloom and light availability is clearly indicated, but I'm afraid that the reader may not catch what the authors would like to show in MYI diagram. Please modify the MYI diagram to show

the relationship of phytoplankton abundance and light availability. Also, the second sentence (How these physical changes. . .) may be omitted from the figure caption. Author's response. We thank the referee for the insight and agree that the figure should be made clearer. Author's changes in manuscript. The following modifications were made to Figure 10. On the first panel (MYI) the arrow (light) going from the sun and through the thicker ice was presented as discontinued (dotted arrow) to signify reduced intensity of light reaching the surface of the water and available for phytoplankton growth. Part of the light is absorbed by the ice (one arrow ending in the ice), and another part of the light is reflected back (2 arrows pointing upwards). On the second panel (FYI) the arrows (light) going from the sun and through the thinner ice and the melt ponds at the surface of the ice show scattering and an increase in the amount of light reaching the surface of the water and available for phytoplankton. Part of the light is absorbed by the ice (one arrow ending in the ice), and another part of the light is reflected back (1 arrow pointing upwards).

Furthermore, as suggested, we modified the caption as follows and took the second sentence out. Initial version: Figure 10: Conspicuous alterations in the Arctic Ocean are underway and include reductions in snow cover, sea ice extent and thickness, and increase in melt pond areal coverage, the occurrence of which is linked to profound modifications in light availability in surface waters below the ice and at its margin. How these physical changes will impact the dynamics of bloom-forming microorganisms and their production of the biogenic climate-active gas DMS are still unknown. The conceptual diagram depicts two types of ice edges (top panel MYI and lower panel FYI) and their potential role in modulation light penetration under the ice pack and the development of phytoplankton blooms and associated DMS dynamics.

Modified version: Figure 10: Conspicuous alterations in the Arctic Ocean are underway and include reductions in snow cover, sea ice extent and thickness, and increase in melt pond areal coverage, the occurrence of which is linked to profound modifications in light availability in surface waters below the ice and at its margin. The conceptual

diagram depicts two types of ice edges (top panel MYI and lower panel FYI) and their potential role in modulation light penetration under the ice pack and the development of phytoplankton blooms and associated DMS dynamics. In this very simplified diagram, reduced light penetration (dotted arrow) and greater light reflection (arrows pointing upwards) occurs in the presence of MYI (top panel) whereas increased light penetration occurs through thinner and ponded FYI (lower panel) allowing phytoplankton to develop under the ice and potentially produce DMS.
* * *
[Figure]

[Figure]

**Fig. 1.** Figure 2. Lizotte et al. Biogeosciences

[Figure]

[Figure]

**Fig. 2.** Figure 10. Lizotte et al. Biogeosciences

---

## Author Comment (AC2) · 4 Feb 2020

Referee #2 Alison Webb a.l.webb@rug.nl Interactive comment on "Phytoplankton and dimethylsulfide dynamics at two contrasting Arctic ice edges" by Martine Lizotte et al.

 This manuscript is a well written and comprehensive review of different parameters determining the dynamics of DMS and DMSP along the ice edges around Greenland and the Canadian arctic. It is a good addition to the currently limited dataset on the relationship between DMS and the sea ice, and highlights the differences between single and multi-year ice. As the Arctic experiences greater ice loss, the importance of the single year ice is going to become more important to the global climate cycle of DMS, but we need to understand the difference

to multi-year ice too. This particular aspect was well covered in the final discussion. I recommend this manuscript for publication with the following minor revisions. Author's response. We thank Dr. Alison Webb for her constructive review of the manuscript. Below we address each point brought up by Dr. Webb.

Specific comments L71. Please include a comment to the potential of Bacterioplankton to produce DMSP (Curson et al 2017) as well as break it down. Author's response. A very good suggestion. Author's changes in manuscript. The following phrase was added from L70 to L72: "The biosynthesis of DMSP is not restricted to eukaryotic organisms, however, and has also been found in marine bacterioplankton who can both produce it and break it down (Curson et al. 2017)." Furthermore, the following reference was added to the reference section on L822: Curson, A., Liu, J., Bermejo Martínez, A., Green, R. T., Chan, Y., Carrión, O., Williams, B. T., Zhang, S.-H., Yang, G.-P., Bulman Page, P. C., Zhang, X.-H and Todd, J. D.: Dimethylsulfoniopropionate biosynthesis in marine bacteria and identification of the key gene in this process, Nat. Microbiol., 2, 17009, https://doi.org/10.1038/nmicrobiol.2017.9, 2017.

L426. 'Bared no significantly' should be 'bore no significant' Author's response. Yes this is correct. Author's changes in manuscript. At L426 (now L436), the word 'bared' was changed to 'bore'.

L577 Reference spell check Stefels and Dijkhuizen 1996 Author's response. Yes Dr. Webb is correct, the letter 'n' was missing from Dijkhuizen Author's changes in manuscript. At Line 577 (now L590), "Stefels and Dijkhuize 1996" was changed to "Stefels and Dijkhuizen 1996".

Fig 2 caption. Spell check Pannel to panel Author's response. Yes, there is indeed a spelling issue here. Author's changes in manuscript. In the following caption the word "Pannel" was changed to "panel". Figure 2: Ice charts adapted from the Canadian Ice Services (CIS) of Environment Canada showing the presence of ice edges in (a) Lancaster Sound with 9-10/10 ice concentrations (> 15 cm) extending from Devon Island

into the Prince Regent Inlet between Somerset and Baffin islands (July 22, 2014); and in (b) Nares Strait with 9-10/10 ice concentrations near Petermann Glacier (August 1, 2014); panel (c) shows the presence of MYI (2 to 5+ years) at the entrance of Robeson Channel as well as a band of FYI (1) (Ease-Grid Sea Ice Age, Version 3 data set, Tschudi et al. (2016)). See the Canadian Ice Services Archives website for more details about sea ice characteristics. Note that Nares Strait includes Smith Sound, Kane Basin, Kennedy Channel, Hall Basin and Robeson Channel.

Table 1 and 2 captions. Both these captions are the same, despite different parameters highlighted in the tables. Can the captions be made more unique to describe the data in each table? Author's response. The two captions are not the same but they are similar. They do describe different variables: physicochemical characteristics (Table 1) and biogeochemical characteristics (Table 2), but also present the same inherent structure, reason why several elements are repeated. Author's changes in manuscript. The caption for Table 2 was modified from its original version to include a more detailed description of the biogeochemical characteristics found in the Table per se.

Table 2: Biogeochemical characteristics (including concentrations of DMSPt and DMS) of the surface waters sampled during the ArcticNet/NETCARE campaign during July-August 2014 grouped within different regions of the Arctic. Chlorophyll a is noted as Chl a and values in italic between parentheses represent in vivo fluorescence. Percentage of dominant phytoplankton taxa is shown for stations where it was available. Ice-covered stations are marked with an asterix*. Stations in bold are included in the vertical cross section transect figures (Figs. 4 to 7). Values that were not available are noted as 'n.a.'

Additional changes proposed by Lizotte and co-authors The authors would like to thank the referees for their review of the paper. Some modifications were made to the Acknowledgements section to reflect this.

Original version (Lines 670 to 675). The authors wish to thank the captain of the

CCGS Amundsen, Alain Lacerte, as well as his officers and crew for their support during the oceanographic campaign. We also want to thank Jonathan Gagnon for nutrient analysis, and Sylvie Lessard for the taxonomic analysis. This study received financial support from NETCARE (funded under the NSERC Climate Change and Atmospheric Research program), ArcticNet (Network of Centres of Excellence of Canada) and the NSERC Discovery Grant Program and Northern Research Supplement Program (M. Levasseur, M. Gosselin and J.-É. Tremblay). This paper is a contribution to the research programmes of NETCARE, ArcticNet, and Québec-Océan.

Updated version (Line 684 to 691) The authors wish to thank the captain of the CCGS Amundsen, Alain Lacerte, as well as his officers and crew for their support during the oceanographic campaign. We also want to thank Jonathan Gagnon for nutrient analysis, and Sylvie Lessard for the taxonomic analysis. Finally the authors acknowledge both Dr Alison Webb and an anonymous referee for their constructive reviews of the original manuscript. This study received financial support from NETCARE (funded under the NSERC Climate Change and Atmospheric Research program), ArcticNet (Network of Centres of Excellence of Canada) and the NSERC Discovery Grant Program and Northern Research Supplement Program (M. Levasseur, M. Gosselin and J.-É. Tremblay). This paper is a contribution to the research programmes of NETCARE, ArcticNet, and Québec-Océan.